# Bridging Lottery Ticket and Grokking:
# Understanding Grokking from Inner Structure of Networks

**Gouki Minegishi**     **Yusuke Iwasawa**     **Yutaka Matsuo**
*{minegishi,iwasawa,matsuo}@weblab.t.u-tokyo.ac.jp*
*The University of Tokyo*

**Reviewed on OpenReview:** *https: // openreview. net/ forum? id= eQeYyup1tm*

## Abstract

Grokking is an intriguing phenomenon of *delayed generalization*, where neural networks initially memorize training data with perfect accuracy but exhibit poor generalization, subsequently transitioning to a generalizing solution with continued training. While factors such as weight norms and sparsity have been proposed to explain this delayed generalization, **the influence of network structure** remains underexplored. In this work, we link the grokking phenomenon to the lottery ticket hypothesis to investigate the impact of internal network structures. We demonstrate that utilizing lottery tickets obtained during the generalizing phase (termed *grokked tickets*) significantly reduces delayed generalization across various tasks, including multiple modular arithmetic operations, polynomial regression, sparse parity, and MNIST classification. Through controlled experiments, we show that the mitigation of delayed generalization is not due solely to reduced weight norms or increased sparsity, but rather to the discovery of good subnetworks. Furthermore, we find that grokked tickets exhibit periodic weight patterns, beneficial graph properties such as increased average path lengths and reduced clustering coefficients, and undergo rapid structural changes that coincide with improvements in generalization. Additionally, pruning techniques like the edge-popup algorithm can identify these effective structures without modifying the weights, thereby transforming memorizing networks into generalizing ones. These results underscore the novel insight that structural exploration plays a pivotal role in understanding grokking.

## 1 Introduction

Understanding the mechanism of generalization is a central question in understanding the efficacy of neural networks. Recently, Power et al. (2022) unveiled the intriguing phenomenon of *delayed generalization (grokking)*; neural networks initially attain a *memorizing network* $C_{\mathrm{mem}}$ with the perfect training accuracy but poor generalization, yet further training transitions the solution to a *generalizing network* $C_{\mathrm{gen}}$. This phenomenon, which contradicts standard machine learning expectations, is being studied to answer the question: *what underlies the transition between memorization and generalization?* (Liu et al., 2022; 2023a)

Regarding the relationship between generalization and deep learning in general, it is well known that *structure of networks* significantly impacts generalization performance. For instance, image recognition performance has greatly improved by leveraging the structure of convolution (Krizhevsky et al., 2012). Moreover, as shown in Neyshabur (2020), incorporating $\beta$-Lasso regularization into fully connected MLPs facilitates the emergence of locality — resembling the structures in CNNs — leading to improved performance in image tasks. From a slightly different perspective, Frankle & Carbin (2019) proposed the lottery ticket hypothesis (LTH), which suggests that good subnetworks (good structure) help to achieve better performance with better sample efficiency (Zhang et al., 2021). Also, Ramanujan et al. (2020) shows that exploring structures alone can achieve performance comparable to weight updates, suggesting that good subnetworks are enough to achieve generalized performance.

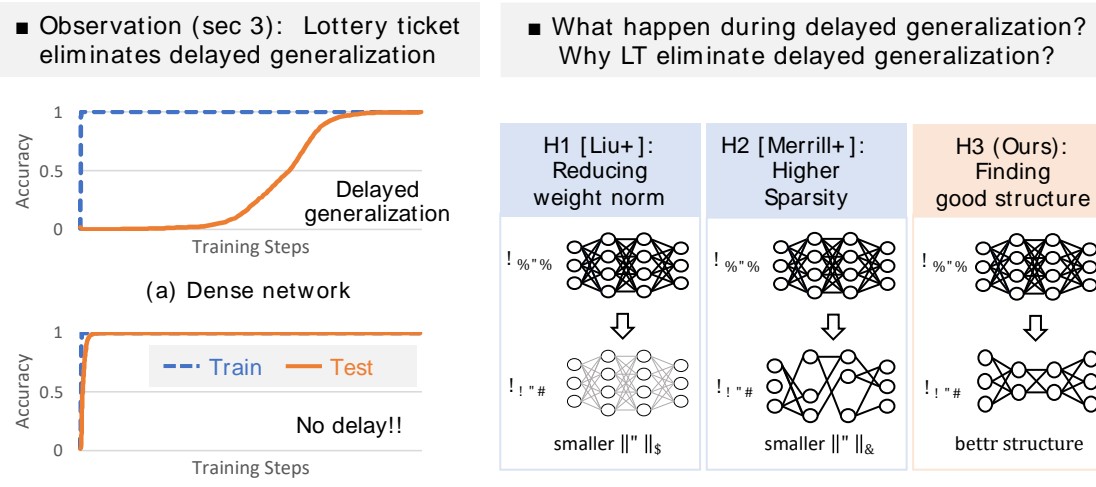

Figure 1: (Left) Accuracy of dense model and the lottery ticket obtained at generalizing solution (grokked ticket). When using a lottery ticket (good subnetworks), the train and test accuracy increase almost similarly, i.e., the time from memorization ($t_{\mathrm{mem}}$) to generalization ($t_{\mathrm{gen}}$) has significantly accelerated. Note that not only the subtraction ($t_{\mathrm{gen}} - t_{\mathrm{mem}}$) but the ratio ($t_{\mathrm{gen}}/t_{\mathrm{mem}}$) is also significantly improved, meaning that it's not just a matter of faster learning. (Right) Three hypotheses on why delayed generalization is reduced with a lottery ticket. We show that it is not due to a reduction in weight norm or an increase in sparsity, but rather the discovery of *good structure*.

While the importance of structure is well known in general, its connection to the phenomenon of grokking has not been investigated enough. Similar to our study, several prior works connect the grokking phenomenon to the property of networks, e.g., weight norm and sparsity of networks. For example, Liu et al. (2023a) experimentally confirmed that generalizing solutions have smaller norms compared to memorizing solutions. The original paper (Power et al., 2022) showed that adding weight decay during training is necessary for triggering grokking. However, our work differs from these works by discussing the relationship between the process of discovering good structures (i.e., subnetworks) and grokking rather than merely reducing the weight norm.

To investigate the role of structure in the grokking phenomenon, we first demonstrate that when using the lottery ticket obtained at generalizing solution (referred to as grokked tickets), delayed generalization is significantly reduced. Figure 1 (left) illustrates that the train and test accuracy increase almost simultaneously with grokked tickets, unlike randomly initialized dense networks where delayed generalization occurs. As will be shown later, this result is related to the pruning rate, with proper pruning rates resulting in less delay. We conducted further experiments from several perspectives to understand why delayed generalization is significantly reduced with grokked tickets. First, as illustrated in Figure 1 (right), we decompose the differences between the grokked ticket and a randomly initialized dense network into three elements: (1) small weight norms, (2) sparsity and (3) good structure. We investigate which of these elements contribute to the significant reduction of delayed generalization. For (1) weight norms, we find that dense networks with the same initial weight norms as the grokked ticket do not generalize faster, indicating weight norm is *not* the cause. For (2) sparsity, networks with the same parameter size using well-known pruning methods (Wang et al., 2019; Lee et al., 2019; Tanaka et al., 2020) also do not generalize faster, indicating sparsity is *not* the cause. These results suggest that (3) good structure is essential for understanding grokking.

Building on these findings, we delve deeper into understanding the nature and role of these structural properties in the grokking phenomenon. Specifically, we investigate several key aspects. First, to determine whether the acquisition of the structure is synchronized with improvements in generalization performance, we use the Jaccard distance (Jaccard, 1901) as a metric of structural distance. Our analysis shows that the structure of the subnetwork changes rapidly during the transition from memorization to generalization. This rapid structural adaptation is synchronized with enhancements in generalization performance. Second, we

examine both graph-theoretic properties, including increased average path lengths and reduced clustering coefficients, and the periodic nature of the Modular Addition task (Nanda et al., 2023). We find that beneficial graph properties emerge and align closely with improved generalization, and the model exploits the task's inherent periodicity to discover internally well-structured subnetworks suited for achieving better generalization. Finally, based on our results indicating that structure exploration is crucial for grokking, we demonstrate that pruning facilitates generalization. We employ the edge-popup algorithm (Ramanujan et al., 2020) to identify a good structure while keeping the weights unchanged. Our experiments show that the memorizing network can be transferred to a generalizing network through pruning without weight updates, thereby strengthening our claim that delayed generalization occurs due to the discovery of a good structure.

In summary, our contributions are below:

- **Linking Lottery Tickets and Grokking** (section 3): We investigate the role of inner structures (subnetworks) in the grokking process by connecting lottery tickets to delayed generalization. Our results demonstrate that using lottery tickets significantly reduces the delayed generalization, highlighting the importance of subnetworks in achieving efficient generalization.

- **Decoupling Properties of Lottery Tickets** (section 4): We systematically separate the intrinsic properties of lottery tickets into three components: (1) weight norm, (2) sparsity, and (3) good structure. Through a series of controlled experiments, we show that neither weight norm nor sparsity alone accounts for the reduction in delayed generalization. Instead, it is the discovery of a good structural configuration within the network that is crucial.

- **Understanding the Characteristics of Good Structure** (section 5): We delve deeper into the nature of the beneficial structures discovered by lottery tickets. Our findings reveal that grokked tickets exhibit periodic weight representations and undergo rapid structural changes synchronized with improvements in generalization performance. Furthermore, we demonstrate that pruning techniques, such as the edge-popup algorithm, can identify these effective structures without altering the weights, effectively transitioning memorizing networks into generalizing ones. This highlights the pivotal role of structural exploration in mitigating delayed generalization.

## 2 Background

**Grokking** is a phenomenon where generalization happens long after overfitting the training data (as shown in Figure 1 (left)). The phenomenon was initially observed in the modular addition task ($(a + b)$ mod $p$ for $a, b \in (0, \cdots, p - 1)$), and the same phenomenon has been observed in more complex datasets, encompassing modular arithmetic (Gromov, 2023; Davies et al., 2023; Rubin et al., 2023; Stander et al., 2023; Furuta et al., 2024), semantic analysis (Liu et al., 2023a), n-k parity (Merrill et al., 2023), polynomial regression (Kumar et al., 2023), hierarchical task (Murty et al., 2023) and image classification (Liu et al., 2023a; Radhakrishnan et al., 2022). This paper mainly focuses on grokking in the modular arithmetic tasks commonly used in prior studies.

To understand grokking, previous works proposed possible explanations, including the slingshot mechanism (Thilak et al., 2022), random walk among minimizers (Millidge, 2022), formulation of good representation (Liu et al., 2022), the scale of weight norm (Liu et al., 2023a; Varma et al., 2023), simplicity of Fourier features (Nanda et al., 2023) and sparsity of generalizing network (Miller et al., 2023).

Among those, one of the dominant explanations regarding how the network changes during the process of grokking are the simplicity of the generalization solution, particularly focusing on the weight norms of network parameters $\|\boldsymbol{\theta}\|_2$. For example, the original paper (Power et al., 2022) posited that weight decay plays a pivotal role in grokking, i.e., test accuracy will not increase without weight decay. Liu et al. (2023a) analyzed the loss landscapes of train and test dataset, verifying that grokking occurs by entering the generalization zone defined by L2 norm, with models having large initial values $\boldsymbol{\theta}_0$. More recently, Varma et al. (2023) demonstrated that the generalization solution could produce higher logits with smaller weight norms. In this paper, we examine the changes in the network's structure and demonstrate that the network is not simply decreasing its overall weight norms but searching for good structures within itself.

Several studies have proposed that acquiring good representations is the key to understanding grokking. For example, Power et al. (2022); Liu et al. (2022) explained that the topology of the ideal embeddings tends to be circles or cylinders within the context of modular addition tasks. Nanda et al. (2023) identified the trigonometric algorithm by which the networks solve modular addition after grokking and showed that it grows smoothly over training. Gromov (2023) showed an analytic solution for the representations when learning modular addition with MLP. Zhong et al. (2023) show, using modular addition as a prototypical problem, that algorithm discovery in neural networks is sometimes more complex. These studies support the quality of representation as key to distinguishing memorizing and generalizing networks; however, these studies do not explain what is happening within the network's structure.

The **lottery ticket hypothesis** proposed by Frankle & Carbin (2019) has garnered attention as an explanation for why over-parameterized neural networks exhibit generalization capabilities (Allen-Zhu et al., 2019). Informally, the lottery ticket hypothesis states that randomly initialized over-parameterized networks include sparse subnetworks that reach good performance after train, and the existence of the subnetworks is key to achieving good generalization in deep neural networks. This claim was initially demonstrated experimentally, but theoretical foundations have also been established (Frankle et al., 2020; Sakamoto & Sato, 2022). More formally, the process involves the following steps:

1. Initialize dense network $f_{\boldsymbol{\theta_0}}$ and train the network for $t$ epochs to obtain the weights $\boldsymbol{\theta_t}$

2. Perform $k\%$ pruning on the trained network based on absolute values $|\boldsymbol{\theta_t}|$. This process, known as magnitude pruning, yields a mask $\boldsymbol{m}_t^k \in \{0,1\}^{|\boldsymbol{\theta_t}|}$.

3. Reset the weights of the network to their initial values. $\boldsymbol{\theta_0}$ and get a subnetwork $f_{\boldsymbol{\theta_0} \odot \boldsymbol{m}_t^k}$, representing *lottery ticket*. Train the subnetwork for $t'$ epochs and obtain $f_{\boldsymbol{\theta'_{t'}} \odot \boldsymbol{m}_t^k}$.

After the discovery of the lottery tickets, Ramanujan et al. (2020) show that there exist **strong lottery tickets**, which achieve good performance without weight update. They use the **edge-popup** algorithm (Ramanujan et al., 2020) to selects subnetworks based on a score $\boldsymbol{s}$ ($\boldsymbol{s} \in \mathbb{R}^{|\boldsymbol{\theta_0}|}$). In other words, when pruning a certain proportion $k$ of weights from the given weights $\boldsymbol{\theta_0}$, the model predicts using edges with the top $(1-k)$ scores in a forward pass. For a detailed description of edge-popup, refer to Appendix A. In section 5, we use the edge-popup algorithm to check if pruning can be worked as a force to accelerate the grokking process.

## 3 Lottery Tickets significantly reduces Delayed Generalization

### 3.1 Experiment Setup

Following Power et al. (2022) and other grokking literatures (Nakkiran et al., 2019; Liu et al., 2022; Gromov, 2023; Liu et al., 2023a), we constructed a dataset of equations of the form: $(a+b)\%p = c$. The task involves predicting $c$ given a pair of $a$ and $b$. Our setup uses the following detailed configurations: $p = 67$, $0 \leq a, b, c < p$. The dataset size is 2211, considering all possible pairs where $a \geq b$. We split it into training (40%) and test (60%) following Liu et al. (2022).

Following Liu et al. (2022), we design the MLP as follows. Firstly, we map the one-hot encoding of $\boldsymbol{a}$, $\boldsymbol{b}$ with the embedding weights $W_{\text{emb}}$: $\boldsymbol{E}_a = W_{\text{emb}}\boldsymbol{a}$, $\boldsymbol{E}_b = W_{\text{emb}}\boldsymbol{b}$. We then feed the embeddings $\boldsymbol{E}_a$ and $\boldsymbol{E}_b$ into the MLP as follows:

$$\text{softmax}(\sigma((\boldsymbol{E}_a + \boldsymbol{E}_b)W_{\text{in}})W_{\text{out}}W_{\text{unemb}}) \tag{1}$$

where $W_{\text{emb}}$, $W_{\text{in}}$, $W_{\text{out}}$, and $W_{\text{unemb}}$ are the trainable parameters, and $\sigma$ is an activation function ReLU (Nair & Hinton, 2010). The dimension of the embedding space is 500, and $W_{\text{in}}$ projects into 48-dimensional neurons.

Following Nanda et al. (2023), we used the AdamW optimizer (Loshchilov & Hutter, 2019) with a learning rate $10^{-3}$, the weighting of weight decay $\alpha = 1.0$, $\beta_1 = 0.9$, and $\beta_2 = 0.98$. We initialize weights as $\boldsymbol{\theta_0} \sim \mathcal{N}(0, \kappa/\sqrt{d_{\text{in}}})$, where $d_{\text{in}}$ represents the dimensionality of the layer preceding each weight. If nothing is specified, assume $\kappa = 1$. Let us assume we have training datasets $\mathbf{S}_{\text{train}}$ and test datasets $\mathbf{S}_{\text{test}}$, and train a neural network $f(\boldsymbol{x}; \boldsymbol{\theta})$ where $\boldsymbol{x}$ is an input and $\boldsymbol{\theta}$ represents weight parameters of the networks. Specifically,

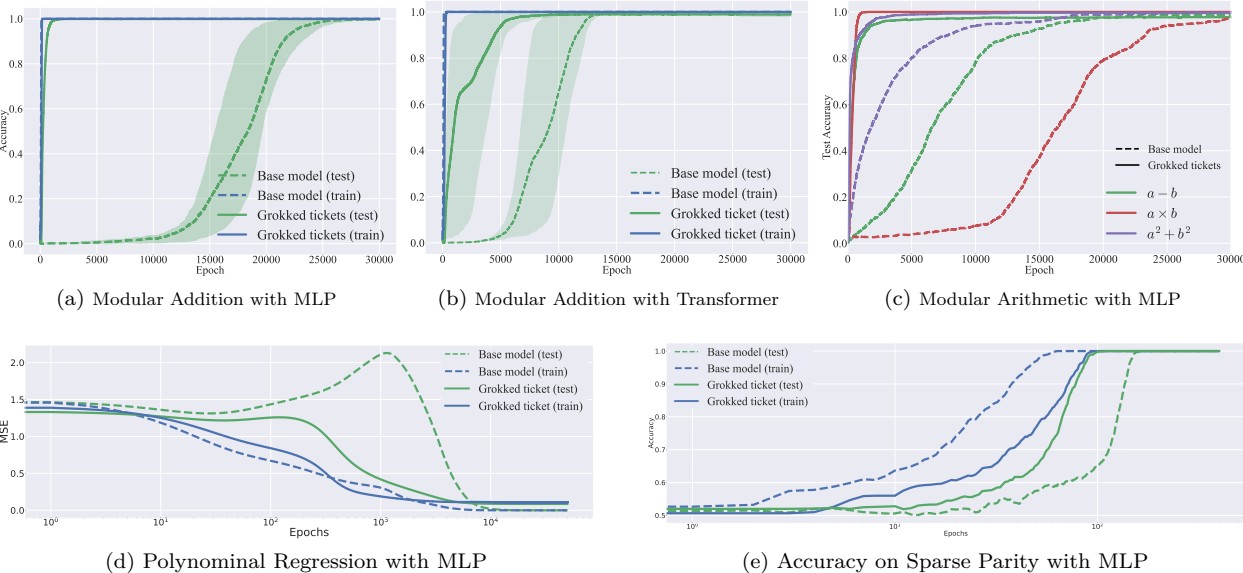

(a) Modular Addition with MLP  (b) Modular Addition with Transformer  (c) Modular Arithmetic with MLP

(d) Polynominal Regression with MLP    (e) Accuracy on Sparse Parity with MLP

Figure 2: Comparing the grokking speed of dense networks and grokked tickets on various setups. (a) Modular addition with MLP, (b) Modular addition with Transformer, and (c) Other modular arithmetic tasks (represented by color) and experiments other than modular arithmetic: (d) loss on polynomial regression, (e) accuracy on sparse parity. The dashed line represents the accuracy of the base model, and the solid line represents that of grokked tickets. In all setups, the time to generalization ($t_{\text{gen}}$) is reduced by grokked tickets.

the network is trained using AdamW over a cross-entropy loss and weight decay (L2 norm of weights $\|\boldsymbol{\theta}\|_2$):

$$\operatorname*{argmin}_{\boldsymbol{\theta}} \mathbb{E}_{(\boldsymbol{x},y)\sim S}\left[\mathcal{L}(f(\boldsymbol{x};\boldsymbol{\theta}),y) + \frac{\alpha}{2}\|\boldsymbol{\theta}\|_2\right].$$

To quantitatively measure how much-delayed generalization is reduced, we define $t_{\text{mem}}$ as the step at which the training accuracy exceeds $P\%$, and $t_{\text{gen}}$ as the step at which the test accuracy exceeds $P\%$. Following Kumar et al. (2024), we use $P = 95$ for modular arithmetic tasks. We use the proposition ($\tau_{\text{grok}} = t_{\text{gen}}/t_{\text{mem}}$) to measure the acceleration.

We compared the performance of 1) dense networks $f_{\boldsymbol{\theta}_{t'}}$ and 2) trained lottery tickets $f_{\boldsymbol{\theta}'_{t'} \odot \boldsymbol{m}_t^k}$, where $t'$ is a training epoch to get the final score, $t$ is timing of pruning, and $k$ is a pruning ratio. As a special case, when $t \geq t_{\text{gen}}$, we denote the subnetworks as **grokked tickets**. We tested various $t$ and $k$ and investigated how they change the generalization speed. By default, we used $k = 0.6$.

## 3.2 Results

Figure 2-(a) shows the test accuracy of the grokked ticket and the base model on the modular addition task. The base model refers to a dense model trained from random initial values. The grokked ticket shows an improvement in test accuracy at nearly the same time as the improvement in training accuracy of the base model. In Figure 2-(b), using experiments with transformers, the result also shows that grokked tickets result in less delay of generalization. Following Power et al. (2022), we conducted experiments on various modular arithmetic tasks to demonstrate the elimination of delayed generalization. Figure 2-(c) shows a comparison of the base model (dashed line) and grokked ticket (solid line) with various modular arithmetic tasks. Moreover, following Kumar et al. (2023); Pearce et al. (2023), we also demonstrate that delayed generalization is reduced by the grokked ticket in both the polynomial regression and sparse parity tasks in Figure 2-(d,e). The results show that grokked tickets significantly reduced delayed generalization even on various tasks. See Appendix B for a detailed explanation of the experimental setup.

Figure 3-(a) quantify the relationship between pruning rate and $\tau_{\text{grok}} = t_{\text{gen}}/t_{\text{mem}}$ (log-scaled). When the pruning rate is 0.7, $\tau_{\text{grok}}$ reaches its minimum, indicating that grokked ticket significantly reduce delayed

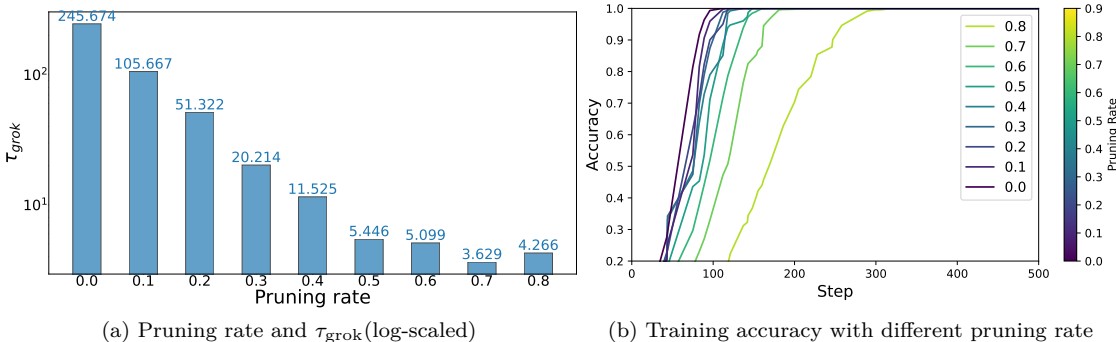

(a) Pruning rate and $\tau_{\text{grok}}$ (log-scaled)

(b) Training accuracy with different pruning rate

Figure 3: Quantitative comparison of grokking speed among different pruning rates. Note that pruning rate = 0.0 corresponds to the dense network. The definition of the $\tau_{\text{grok}}$ is explained in subsection 3.1

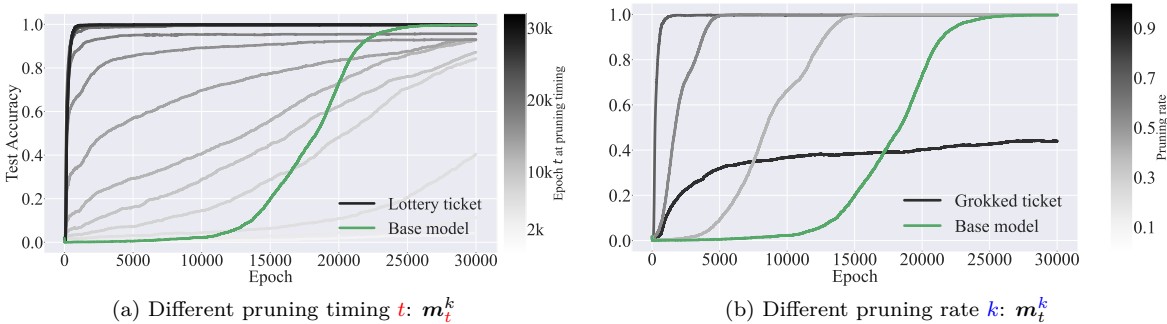

(a) Different pruning timing $t$: $\boldsymbol{m}_t^k$

(b) Different pruning rate $k$: $\boldsymbol{m}_t^k$

Figure 4: (a) Comparison of the test accuracy of different epoch $t$ in which lottery tickets are acquired. We conducted every 2k epochs. The lottery tickets obtained before 25k epochs (non-grokked tickets) do not fully generalize. Additionally, this generalization ability corresponds to the test accuracy of the base model. The lottery tickets obtained after 25k epochs (grokked tickets) reduced delayed generalization. (b) The effect of pruning rate $k$ on grokked tickets. We conducted every 0.2 pruning rate. Most pruning ratios (0.1, 0.3, 0.5, and 0.7) accelerate the generalization, indicating that the above observation does not depend heavily on the selection of the pruning ratio.

generalization. Note that, as shown in Figure 3-(b), the $t_{\text{mem}}$ is *delayed* when using a higher pruning rate, meaning that the grokked ticket are not simply accelerating the entire learning process but specifically speeding up the transition from memorization to generalization.

In Figure 4-(a), we compare the test accuracy of different epoch $t$ in which lottery tickets are acquired. The results show that lottery tickets obtained before 25k epochs (non-grokked tickets) do not fully generalize. Additionally, this generalization ability corresponds to the test accuracy of the base model. On the other hand, lottery tickets obtained after 25k epochs (grokked tickets) get perfect generalization and reduce delayed generalization. In Figure 4-(b), we investigate the effects of pruning rate in grokked ticket, indicating if it's too large (e.g., 0.9), it can't generalize; if it's too small (e.g., 0.3), it doesn't generalize quickly enough.

## 4 Decoupling Lottery Tickets: Norm, Sparsity, and Structure

In section 3, we established that *grokked tickets* — subnetworks extracted at the onset of generalization — significantly reduce the delayed generalization phenomenon. While these results strongly suggest that some intrinsic property of the grokked tickets is responsible for their rapid move to generalization, it is not yet clear what exactly this property is. Recall from Figure 2 that a grokked ticket can drastically reduce delayed generalization compared to a base model (a densely trained network from scratch). To understand the key factor behind this effect, we consider three hypotheses which are illustrated in Figure 1 (right):

**Hypothesis 1 (Weight Norm):** *Grokked tickets have smaller weight norms than the base model, and this reduction in weight norm leads to the reduction in delayed generalization.*

In Section 4.1, we test Hypothesis 1 by preparing the dense model that matches the weight norms of the grokked ticket. If these models, despite having the same weight norms, exhibit different generalization speeds, it implies that the weight norm alone does not account for the reduction in delayed generalization. Consequently, Hypothesis 1 is rejected.

**Hypothesis 2 (Sparsity):** *Grokked tickets are sparser subnetworks compared to the base model, and this higher degree of sparsity leads to the reduction in delayed generalization.*

In Section 4.2, we examine Hypothesis 2 by constructing models with the same sparsity level as the grokked ticket. If these equally sparse models do not show a reduction in delayed generalization, it indicates that sparsity alone is not responsible for the reduction in delayed generalization. Therefore, Hypothesis 2 is rejected.

**Hypothesis 3 (Good Structure):** *Beyond differences in weight norm or sparsity, grokked tickets possess a superior structural configuration that directly accelerates generalization.*

If both Hypotheses 1 and 2 are rejected, the only remaining explanation is that the *good structure* inherent to grokked tickets drives the reduction in delayed generalization.

### 4.1 Controlling Weight Norm of Initial Network

To test Hypothesis 1, we prepared two dense models with the same L2 and L1 norms as the grokked ticket, named the 'controlled dense model'. Such dense models are obtained through the following process:

1. Obtain lottery tickets after full generalization $\boldsymbol{m}_{t_{\text{gen}}}^k$.

2. Get weight $L_p$ norm ratio $r_p = \frac{\|\boldsymbol{\theta}_0 \odot \boldsymbol{m}_{t_{\text{gen}}}^k\|_p}{\|\boldsymbol{\theta}_0\|_p}$

3. Create weights $\boldsymbol{\theta}_0 \cdot r_p$ with the same $L_p$ norm as the grokked ticket.

Figure 5-(a) shows the test accuracy of the base model, grokked ticket, and controlled dense models. Despite having the same initial weight norms, the grokked ticket arrives at generalization much faster than both controlled dense models. This result indicates that the delayed increase in test accuracy is attributable not to the weight norms but to the discovery of good structure. The left of the Figure 6 shows the dynamics of the L2 norms for each model. Similar to Liu et al. (2023b), L2 norms decrease in correspondence with the rise in test accuracy, converging towards a 'Generalized zone'. However, as shown on the right side of the Figure 6, the final convergence points of the L1 norms vary for each model. This phenomenon of having similar L2 norms but smaller L1 norms suggests that good subnetwork (grokked ticket) weights become stronger, as indicated in Miller et al. (2023). Similar results have also been observed in Transformer (Figure 16). These results demonstrate that the weight norm itself is insufficient to explain grokking, refuting Hypothesis 1.

### 4.2 Controlling Sparsity

Next, we examine Hypothesis 2, which posits that higher sparsity in grokked tickets accounts for their reduced delayed generalization. To investigate this, we compared models with the same sparsity as the grokked ticket, using various pruning-at-initialization (PaI) methods. Specifically, we tested three well-known PaI methods — Grasp (Wang et al., 2020), SNIP (Lee et al., 2019), and Synflow (Tanaka et al., 2020)) — alongside random pruning as baseline methods. For details on each of the pruning methods, refer to the section Appendix D. Figure 5-(b) compare the transition of the test accuracy of these PaI methods and the grokked ticket. The results show that all PaI methods perform worse than the base model or, in some cases, perform worse than the random pruning. These results indicate that mere sparsity cannot account for the reduced delayed generalization, thereby rejecting Hypothesis 2.

The rejection of Hypothesis 1 and 2, this leaves only Hypothesis 3: the key difference must lie in the good structure of grokked tickets, enabling them to achieve rapid generalization and effectively shorten the delayed generalization period.

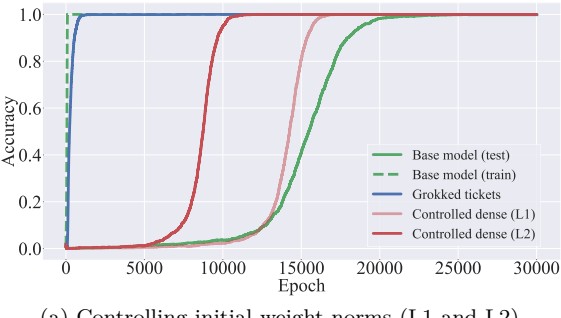

(a) Controlling initial weight norms (L1 and L2)

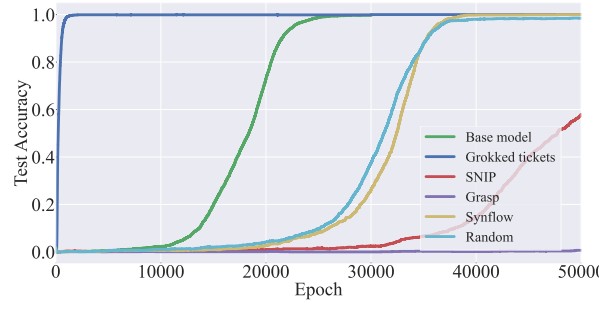

(b) Controlling sparsity w/ pruning at initialization

Figure 5: (a) Test accuracy dynamics of the base model, grokked ticket, and controlled dense model (L1 norm and L2 norm). The grokked ticket reaches generalization much faster than other models. (b) Comparing test accuracy of the different pruning methods. All PaI methods perform worse than the base model or, in some cases, perform worse than the random pruning. These results indicate neither the weight norm *nor* the sparsity alone is the cause of delayed generalization.

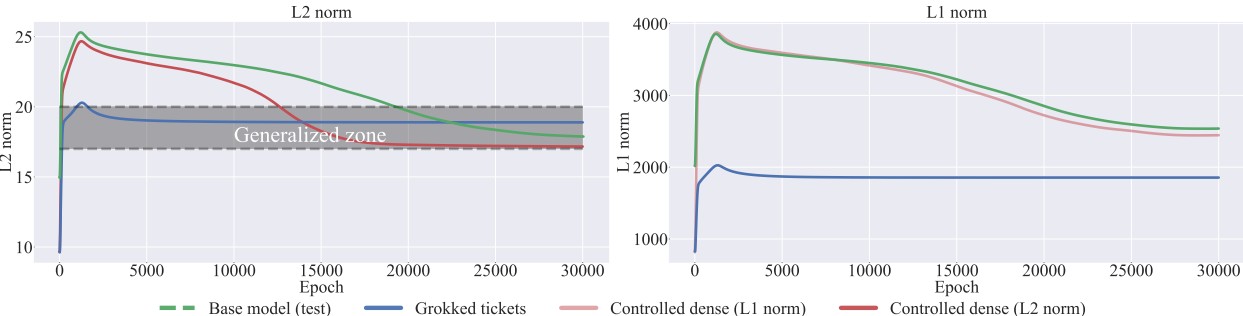

Figure 6: (Left) L2 norm dynamics of the base model, grokked ticket, and controlled dense model. (Right) L1 norm dynamics of the base model, grokked ticket, and controlled dense models. From the perspective of the L2 norm, all models appear to converge to a similar solution (Generalized zone). However, from the perspective of L1 norms, they converge to different values.

## 5 Understanding Grokking from Inner Structure of Networks

In section 4, we demonstrated that weight norm and sparsity are insufficient explanations for delayed generalization. Instead, we suggest that a good structure is essential for understanding grokking. Building on these results, this section delves deeper into understanding the nature and role of the structural properties in grokking. Specifically, we investigate the following aspects:

**Q1 Is the acquisition of structure synchronized with the improvement in generalization performance?** (subsection 5.1)

**Q2 What exactly constitutes a good structure?** (subsection 5.2)

**Q3 Can generalization be achieved solely through structural exploration (pruning) without weight updates?** (subsection 5.3)

### 5.1 Progress Measure: Structural Shift Capture the Generalization Timing

In this section, we conduct a more rigorous analysis of how the good structure is acquired, and we show that the discovery of the good structure corresponds to an improvement in test accuracy.

Firstly, we propose a metric of structural changes in the network, named *Jaccard Distance* (JD) using approach Paganini & Forde (2020); Jaccard (1901). We measure the jaccard distance between the mask at

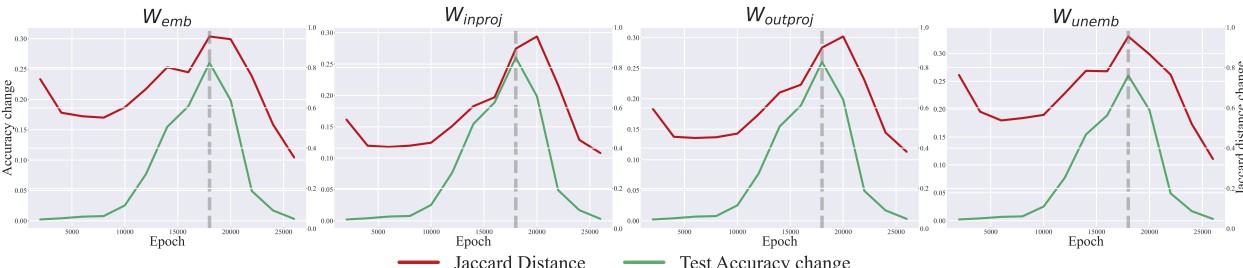

Figure 7: Comparison of jaccard distance (red) and changes in test accuracy (green) on each layer. The jaccard distance is represented as $\mathrm{JD}(t + \delta t, t)$, and the test accuracy change is the difference between epoch $t + \delta t$ and $t$. The vertical line marks the most drastic change in test accuracy. When there is a significant change in test accuracy, the jaccard distance (structural change) increases rapidly.

epoch $t + \delta t$ and $t$.

$$\mathrm{JD}(\boldsymbol{m}_{t+\delta t}, \boldsymbol{m}_t) = 1 - \frac{|\boldsymbol{m}_{t+\delta t} \cap \boldsymbol{m}_t|}{|\boldsymbol{m}_{t+\delta t} \cup \boldsymbol{m}_t|}$$

The $\boldsymbol{m_t}$ represents a mask obtained at $t$ epoch via magnitude pruning and $\delta t$ is 2k epoch. If the two structures differ, this metric is close to 1 and vice versa. Test accuracy change is also represented as a difference between test accuracy at epoch $t + \delta t$ and $t$. In Figure 7, the red line represents the results of the changes in test accuracy and jaccard distance between the mask at epoch $t + \delta t$ and $t$ on each layer. The results show that during significant changes in test accuracy (16k-20k), the maximum change in the mask corresponds, indicating that the discovery of good structure corresponds to an improvement in test accuracy. In the Appendix F, we demonstrate that similar results are obtained for both the polynomial regression and sparse parity tasks.

> **Answer to Q1**
>
> **Is the acquisition of structure synchronized with the improvement in generalization performance?**
>
> **Yes,** the discovery of good structure, as measured by Jaccard Distance, occurs simultaneously with test accuracy improvements, indicating that structural changes drive generalization.

## 5.2 Analysis of Good Structures Through Periodic Structures and Graph Properties

In the previous subsection (subsection 5.1), we established that significant changes in the network's structure, as measured by Jaccard Distance, coincide precisely with improvements in test accuracy. Building on this finding, we now delve deeper into the structural properties that trigger such abrupt differences. Specifically, we investigate the nature of the discovered structure from two different perspectives: (1) the periodic representations known to emerge in modular arithmetic tasks (as studied by Pearce et al. (2023); Nanda et al. (2023)), and (2) graph-theoretic properties that reveal how the network's connectivity evolves to support better generalization. By examining both viewpoints, we uncover how the network ultimately settles on a 'good structure' that drives high test accuracy.

**Modular Addition and Periodicity** Nanda et al. (2023) demonstrated that transformers trained on modular addition tasks rely on these periodic representations to achieve generalization. Specifically, the networks embed inputs $a$ and $b$ into a Fourier basis, encoding them as sine and cosine components of key frequencies $w_k = \frac{2\pi k}{p}$ for some $k \in \mathbb{N}$. These periodic representations are then combined using trigonometric identities within the network layers to compute the modular sum.

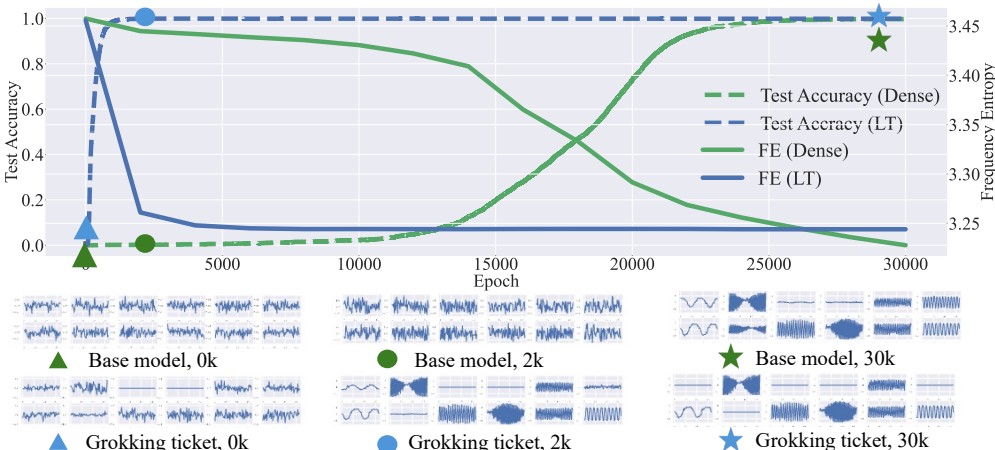

Figure 8: (top) Frequency entropy (FE) and test accuracy of the base model and the grokked ticket. The grokked ticket converges to a smaller frequency entropy much faster than the base model. (bottom) The transition of the input-side weights for each neuron of the base model and grokked ticket. The marks correspond to the epochs of the marks in FE dynamics. The results indicate that grokked tickets acquire good structures for generalization as periodic structures.

By reverse-engineering the weights and activations of a one-layer transformer trained on this task, Nanda et al. (2023) found that the model effectively computes:

$$\cos\big(w_k(a+b)\big) = \cos(w_k a)\cos(w_k b) \; - \; \sin(w_k a)\sin(w_k b), \tag{2}$$

$$\sin\big(w_k(a+b)\big) = \sin(w_k a)\cos(w_k b) \; + \; \cos(w_k a)\sin(w_k b). \tag{3}$$

using the embedding matrix and the attention and MLP layers. The logits for each possible output $c$ are then computed by projecting these values via:

$$\cos\big(w_k(a+b-c)\big) \; = \; \cos\big(w_k(a+b)\big)\cos(w_k c) \; + \; \sin\big(w_k(a+b)\big)\sin(w_k c).$$

This approach ensures that the network's output logits exhibit constructive interference at $c \equiv (a+b) \mod p$, while destructive interference suppresses other incorrect values. Given this mechanism, it can be inferred that the model internally utilizes **periodicity**, such as the addition formulas, to perform modular arithmetic.

**Periodic Structure Acquisition in Grokked Tickets**   In this work, building on this prior research, we examine the structure acquired by grokked tickets from the perspective of periodicity. First, we investigate the periodicity of each neuron by plotting the direction of the weight matrix ($W_{\text{inproj}}$), which is obtained by multiplying $W_{\text{emb}}$ and $W_{\text{in}}$ (refer to subsection 3.1). This two-dimensional weight matrix represents the input dimensions (67) on one axis and the number of neurons (48) on the other. Figure 8 (bottom) illustrates the weights (67 dimensions) of each neuron plotted at training checkpoints (0k, 2k, 30k). In the base model at 30k steps, the weights reveal a clear periodicity, whereas at 0k and 2k steps, the weights exhibit a random structure. Notably, grokked tickets develop this periodic structure much earlier than the base model (see the grokked ticket at **2k** steps). This finding highlights that the structure of grokked tickets is well-suited for Modular Addition, as they acquire a periodic structure that aligns with the task natures.

To quantify this periodicity as a good structure, we introduce Fourier Entropy (FE) as follows. In general, the discrete Fourier transform $\mathcal{F}(\omega)$ of a function $f(x)$ is defined as follows:

$$\mathcal{F}(\omega) = \sum_{x=0}^{N} f(x)\exp\left(-i\frac{2\pi\omega x}{N}\right)$$

In this case, since we want to know the periodicity of each neuron's weights, $f(x)$ is the weight of the $i$-th neuron on the $j$-th input, and $d$ is the dimension of the input. Then, the Fourier Entropy (FE) is calculated

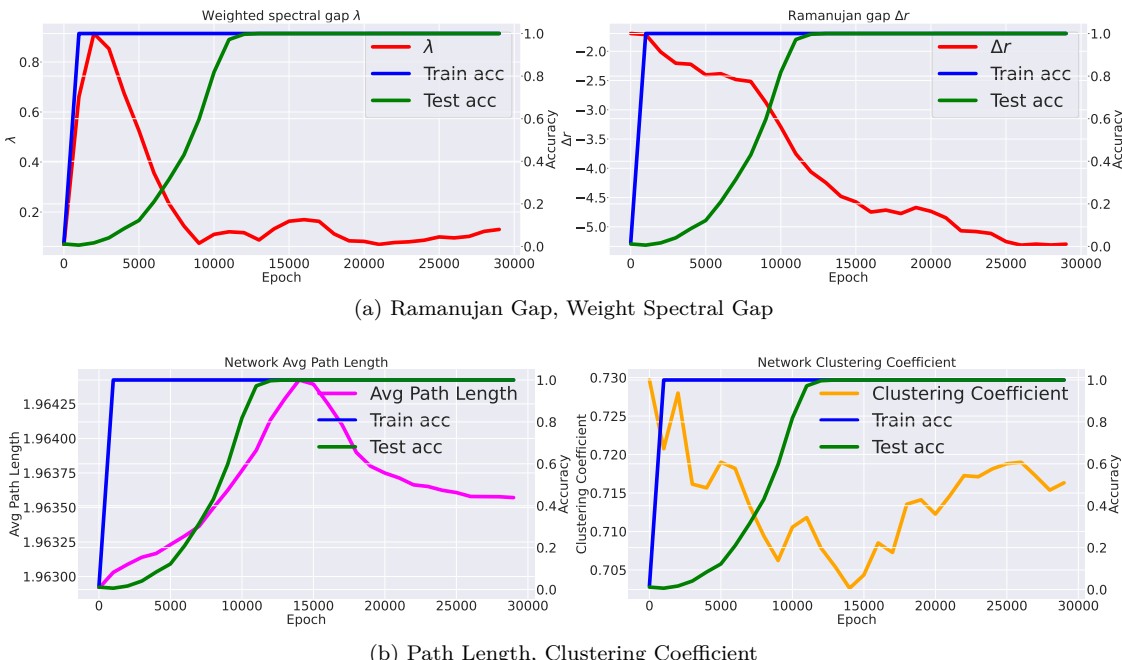

(a) Ramanujan Gap, Weight Spectral Gap

(b) Path Length, Clustering Coefficient

Figure 9: **(a)** Evolution of the weighted spectral gap (left) and Ramanujan gap (right) of the average across the network's layers. **(b)** Average path length (left) and clustering coefficient (right). Definitions and results for individual layers are provided in the Appendix J and Appendix K.

as follows:

$$FE = -\sum_{i=1}^{n} p_i \log p_i$$

Here, $p_i$ is the normalized value of $\mathcal{F}(\omega)$, and $n$ represents a number of neurons. A low value of FE indicates that the **frequency of the weights of each neuron has little variation, converging to specific frequencies**, which shows that the network has acquired periodic structure.

Figure 8 (top) shows the FE of the base model (green) and grokked ticket (blue). The results show that the grokked ticket has neurons with periodic structure at an early stage (2k epochs) and exhibits a rapid decrease in FE in the early epoch. This indicates that the model discovers internally good structures (generalizing structures) suited for the task (Modular Addition). To provide a more detailed analysis, we have added visualizations of the weight matrices and grokked ticket masks in Appendix I.

**Graph Property and Grokked Ticket**   The structures of the network itself exhibit various graph-theoretic properties, as analyzed in prior studies (Hoang et al., 2023b;a; You et al., 2020). Motivated by these insights, we analyze the evolution of the **(1) weighted spectral gap, (2) ramanujan gap, (3) average path length**, and **(4) clustering coefficient** of throughout training, alongside changes in train/test accuracy. Details of each graph property metric are provided in Appendix J.

Figure 9,(a) illustrates how the weighted spectral gap (left) and Ramanujan gap (right) evolve throughout training by examining the average across the network's layers. Results for individual layers are provided in Appendix K. We observe that the weighted spectral gap increases sharply during the memorization phase, indicating a pronounced difference between the largest and second-largest eigenvalues while the model is overfitting. This trend suggests that, in the memorization phase, the weights are dominated by a particular component (corresponding to the leading eigenvector). As the model transitions toward better generalization, however, the spectral gap narrows, reflecting a more balanced utilization of the parameter space. We observe similar behavior in other layers (see Appendix K for details).

Table 1: Test accuracy changes with different optimization methods starting from memorized solutions. WD (Weight Decay) reflects the regularization effect of weight decay, using the same optimization as the base model. In EP w/o WD (Edge-Popup without Weight Decay), accuracy improves solely through pruning, without weight updates. Combining pruning with weight decay in EP w/ WD results in faster generalization than weight decay alone. Additionally, we report the average values of graph property metrics on the network's weight matrix to analyze structural changes over epochs under EP w/o WD.

| Epoch | 600 | 1000 | 1400 | 2000 |
|---|---|---|---|---|
| **Test Accuracy (%)** | | | | |
| WD | $0.53 \pm 0.31$ | $0.95 \pm 0.03$ | $1.00 \pm 0.00$ | $1.00 \pm 0.00$ |
| EP w/o WD | $0.68 \pm 0.19$ | $0.80 \pm 0.17$ | $0.84 \pm 0.16$ | $0.92 \pm 0.06$ |
| EP w/ WD | $\mathbf{0.82 \pm 0.04}$ | $\mathbf{0.96 \pm 0.01}$ | $0.99 \pm 0.00$ | $1.00 \pm 0.00$ |
| **Graph Property Metrics** | | | | |
| Weighted spectral gap | $0.612 \pm 0.05$ | $0.810 \pm 0.11$ | $0.479 \pm 0.04$ | $0.358 \pm 0.03$ |
| Spectral gap | $48.91 \pm 5.05$ | $48.51 \pm 5.12$ | $47.42 \pm 4.97$ | $47.20 \pm 4.55$ |
| Ramanujan gap | $-2.550 \pm 0.52$ | $-4.022 \pm 0.05$ | $-4.335 \pm 0.02$ | $-4.555 \pm 0.01$ |
| Average path length | $1.964 \pm 0.006$ | $1.961 \pm 0.002$ | $1.960 \pm 0.001$ | $1.959 \pm 0.001$ |
| Clustering coefficient | $0.733 \pm 0.41$ | $0.724 \pm 0.35$ | $0.719 \pm 0.30$ | $0.719 \pm 0.28$ |

Figure 9-(b) presents metrics derived from viewing the entire network as a **relational graph** (You et al., 2020). The average path length (left) *increases* alongside the rise in test accuracy, then converges to a moderate range. This trend suggests that when the network is searching for a generalized solution (Grokked ticket), its connectivity becomes more sparse; once that solution is found, the path length decreases and stabilizes. By contrast, the clustering coefficient (right) *decreases* as test accuracy improves, then also settles into an intermediate value. During the phase when the network is searching for a generalized solution (i.e., as the model discovers the Grokked ticket), clustering diminishes but eventually stabilizes near a moderate level. Both metrics thus converge to neither extreme but remain in a balanced regime, consistent with prior studies (You et al., 2020).

---

**Answer to Q2**

**What exactly constitutes a good structure?**

A good structure has low Fourier Entropy, indicating periodic weight patterns aligned with the task. It also follows known graph properties, such as a rising weight spectral gap during memorization and an increasing average path length in generalization.

---

### 5.3 Pruning during Training: Pruning Promote Generalization

Based on the result that the discovery of a good structure corresponds to an improvement in test accuracy, we demonstrate that pruning alone can transition from memorizing solutions to generalizing solutions *without weight update*, and furthermore, the combination of pruning and weight decay promotes generalization more effectively than mere regularization of weight norms.

To verify this, we introduce `edge-popup` (Ramanujan et al., 2020), a method that learns how to prune weights without weight updates. In `edge-popup`, each weight is assigned a score, and these scores are updated through backpropagation to determine which weights to prune. For details regarding `edge-popup`, refer to the Appendix A. We validate our claim by optimizing using three different methods.

**WD** : Training from $\boldsymbol{\theta}_{\mathrm{mem}}$ using **Weight Decay** *with* weight update (same as base model).

**EP w/o WD** : Training from $\boldsymbol{\theta}_{\mathrm{mem}}$ using **Edge-Popup** *without* weight update.

**EP w/ WD** : Training from $\boldsymbol{\theta}_{\mathrm{mem}}$ using **Edge-Popup** and **Weight Decay** *with* weight update.

In Table 1, the result for EP without weight decay (EP w/o WD) shows that the network achieves a generalization performance of 0.92 without any changes to the weights (*merely by pruning weights*). This result demonstrates that generalization can be achieved solely through structural exploration without updating the weights, thereby strengthening our claim that delayed generalization occurs due to the discovery of a good structure. Additionally, the EP w/ WD result shows the fastest improvement in test accuracy and is the most effective in promoting generalization. These insights suggest that practitioners may improve generalization by incorporating methods that directly optimize beneficial structures rather than solely relying on traditional regularization techniques like weight decay. Our findings highlight the potential of grokked tickets to inform the development of new, structure-oriented regularization techniques.

In Table 1, similar to the analysis in subsection 5.2, we examined the average values of graph property metrics on the network's weight matrix under EP w/o WD. As observed in subsection 5.2, the weighted spectral gap initially increased and then declined, while the ramanujan gap continued to decrease. Furthermore, both the average path length and the clustering coefficient converged to similar values.

> **Answer to Q3**
>
> **Can generalization be achieved solely through structural exploration (pruning) without weight updates?**
>
> **Yes,** pruning alone significantly improves generalization, demonstrating that discovering good subnetworks is sufficient to transition from memorization to generalization.

## 6 Discussion and Related Works

In this paper, we conducted a set of experiments to understand the mechanism of grokking (delayed generalization). Below is a summary of observations. (1) In subsection 3.2, the use of the lottery ticket significantly reduces delayed generalization. (2) In section 4, good structure is a more important factor in explaining grokking than the weight norm and sparsity by comparing it with the same weight norm and sparsity level. (3) In subsection 5.2, the structure of grokked tickets is well-suited for task. (4) In subsection 5.1, good structure is gradually discovered, corresponding to improvement of test accuracy. (5) In subsection 5.3, pruning without updating weights from a memorizing solution increases test accuracy. In this section, we connect these results with a prior explanation of the mechanism of grokking and relevant related works.

**Weight norm reduction** Liu et al. (2023a) suggests that the reduction of weight norms is crucial for generalization. In Figure 5, our results go further to show that, rather than simply reducing weight norms, the network discovers good structure (subnetwork), resulting in the reduction of weight norms.

**Sparsity and Efficiency** Merrill et al. (2023) argued that the grokking phase transition corresponds to the emergence of a sparse subnetwork that dominates model predictions. While they empirically studied parse parity tasks where sparsity is evident, we are conducting tasks (modular arithmetic, MNIST) commonly used in grokking research and architecture (MLP, Transformer). Furthermore, Figure 5, we demonstrate not only sparsity but also that good structure is crucial.

**Representation learning** Liu et al. (2022); Nanda et al. (2023); Liu et al. (2023a); Gromov (2023) showed the quality of representation as key to distinguishing memorizing and generalizing networks. Figure 8 demonstrates that good structure contributes to the acquisition of good representation, suggesting the importance of inner structure (network topology) in achieving good representations.

**Regularization** Weight decay (Rumelhart et al., 1986) is one of the most commonly used regularization techniques and is known to be a critical factor in grokking (Power et al., 2022; Liu et al., 2023a). In Appendix H, we show if the good structure is discovered, the network generalizes without weight decay, indicating that weight decay works as a structure exploration, which suggests that weight decay contributes

not to reducing weight but to exploring good structure. In addition, we tested pruning as a new force to induce generalization (Table 1). However, as shown in the results of pruning at initialization (Figure 5-(b)), improper pruning can degrade generalization performance.

**Lottery ticket hypothesis**   The lottery ticket hypothesis (Frankle & Carbin, 2019) suggests that good structures are crucial for generalization, but it remains unclear how these structures are acquired during training and how they correspond to the network's representations. To the best of my knowledge, our paper is the first to connect grokking and the lottery ticket hypothesis, demonstrating how good structures emerge (subsection 5.1) and contribute to effective representations (subsection 5.2). Building on this, we further show that while faster generalization is a well-documented property of winning tickets, our work goes beyond by exploring how the discovery of good structures.

**Future Directions**   While our study focused on simple tasks, the key insight of discovering good subnetworks can extend to more complex domains like large-scale image recognition and language generation. Our findings suggest that good subnetworks, or "lottery tickets", help mitigate delayed generalization, offering a structural perspective on regularization. Traditional methods like weight decay indirectly promote structure discovery, whereas pruning and mask-based optimization provide a more direct and potentially more effective approach. As shown in Table 1, combining weight decay with the edge-popup algorithm accelerates generalization, demonstrating the advantages of structure-aware techniques. Future research could integrate these insights into contemporary architectures to enhance learning efficiency and robustness. Exploring pruning-based or mask-optimization strategies within large-scale models may further validate their effectiveness in real-world applications. Bridging these structural insights with mainstream training techniques could improve model generalization and efficiency in complex settings.

Additionally, based on the insights regarding graph properties obtained in subsection 5.2 of this study, future work should explicitly sample neural network structures according to graph properties, as demonstrated in previous research (Javaheripi et al., 2021). Incorporating the structural insights identified here into the sampling process could facilitate the discovery of more efficient and robust network architectures.

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

## A    Edge-popup algorithm

We provide a detailed explanation of the edge-popup algorithm. Edge-popup Ramanujan et al. (2020) is a method for finding effective subnetworks within randomly initialized neural networks without weight updates.

**Basic Flow:**

1. Random Initialization: Initialize the weights $\mathbf{w}$ of a large neural network randomly.

2. Assign Scores: Assign a score $s_{ij}$ to each edge $w_{ij}$ randomly.

3. Select Subnetwork: Form a subnetwork $\mathcal{G}$ using only the edges with top $k\%$ scores.

4. Optimize Scores: Update the scores $s_{ij}$ based on the performance of the subnetwork $\mathcal{G}$.

**Score Updates:** Update the score $s_{ij}$ of each edge $w_{ij}$ using the following formula:

$$s_{ij} \leftarrow s_{ij} - \alpha \frac{\partial L}{\partial I_j} Z_i w_{ij}$$

- $\alpha$ is the learning rate

- $\frac{\partial L}{\partial I_j}$ is the gradient of the loss with respect to the input of the $j$-th neuron

- $Z_i$ is the output of the $i$-th neuron

**Actual Computation:** Forward Pass: Compute using only the edges whose scores $s_{ij}$ are in the top $k\%$.

$$I_j = \sum_{i \in V} w_{ij} Z_i h(s_{ij})$$

Here, $h(s_{ij})$ is 1 if $s_{ij}$ is in the top $k\%$ and 0 otherwise.

## B  Different configurations of the task and the architecture.

### B.1  MLP for MNIST

We use 4-layer MLP for the MNIST classification. The difference from the regular classification is that we are using Mean Squared Error (MSE) for the loss. We adopted this setting following prior research (Liu et al., 2023a). In (Liu et al., 2023a), it was confirmed in the Appendix that grokking occurred without any problems, even when trying with cross-entropy. Figure 10 shows the test and train accuracy on various configurations. It is evident that grokked tickets accelerate generalization in all configurations, and the exploration of grokked tickets contributes to generalization.

### B.2  Transformer for modular addition

Similar to Nanda et al. (2023), we use a 1-layer transformer in all experiments. We use single-head attention and omit layer norm.

- **Hyperparameters:**
    - $d_{\text{vocab}} = 67$: Size of the input and output spaces (same as $p$).
    - $d_{\text{emb}} = 500$: Embedding size.
    - $d_{\text{mlp}} = 128$: Width of the MLP layer.

- **Parameters:**
    - $W_E$: Embedding layer.
    - $W_{\text{pos}}$: Positional embedding.
    - $W_Q$: Query matrix.
    - $W_K$: Key matrix.
    - $W_V$: Value matrix.
    - $W_O$: Attention output.
    - $W_{\text{in}}$, $b_{\text{in}}$: Weights and bias of the first layer of the MLP.
    - $W_{\text{out}}$, $b_{\text{out}}$: Weights and bias of the second layer of the MLP.
    - $W_U$: Unembedding layer.

We describe the process of obtaining the logits for the single-layer model. Note that the loss is only calculated from the logits on the final token. Let $x_i^{(l)}$ denote the token at position $i$ in layer $l$. Here, $i$ is 0 or 1, as the number of input tokens is 2, and $x_i^{(0)}$ is a one-hot vector. We denote the attention scores as $A$ and the triangular matrix with negative infinite elements as $M$, which is used for causal attention.

The logits are calculated via the following equations:

$$x_i^{(1)} = W_E x_i^{(0)} + W_{\text{pos}} x_i^{(0)},$$
$$A = \text{softmax}(x^{(1)\text{T}} W_K^{\text{T}} W_Q x^{(1)} - M),$$
$$x^{(2)} = W_O W_V (x^{(1)} A) + x^{(1)},$$
$$x^{(3)} = W_{\text{out}} \text{ReLU}(W_{\text{in}} x^{(2)} + b_{\text{in}}) + b_{\text{out}} + x^{(2)},$$
$$\text{logits} = \text{softmax}(W_U x^{(3)}).$$

Figure 10 shows the training and test accuracy of the base model (green) and the grokked ticket (blue). Across datasets (e.g., Modular Addition, MNIST) and architectures (Transformer), the grokked ticket (blue) consistently reaches generalization faster than the base model (green). These results underscore the importance of structural elements in grokking, regardless of the task or architecture.

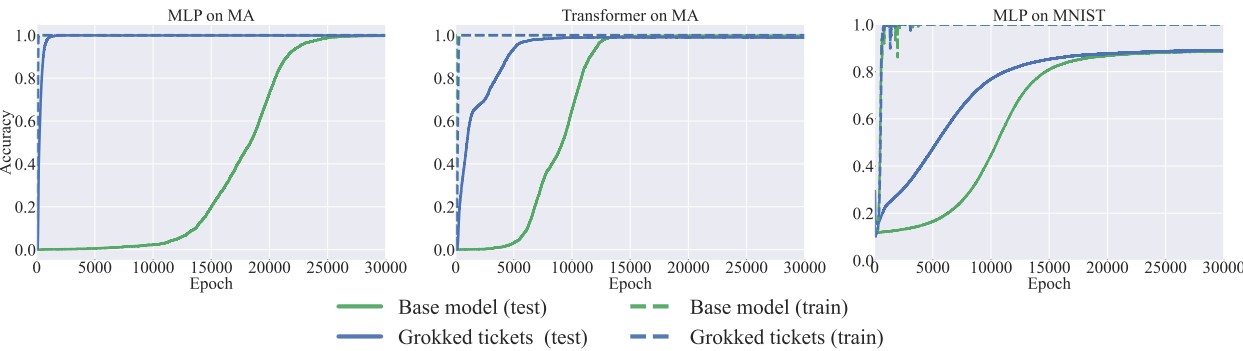

Figure 10: Comparison of base model (green) and grokked ticket (blue). Each column corresponds to the different configurations of the task (Modular Addition and MNIST) and the architecture (MLP and Transformer). The dashed line represents the results of the training data.

## C  Experiments with Different Seeds

To ensure reproducibility, we conduct experiments with three different seeds and present the results for experiments in Figure 2. In addition to accuracy, we plot the loss of train and test in Figure 11.

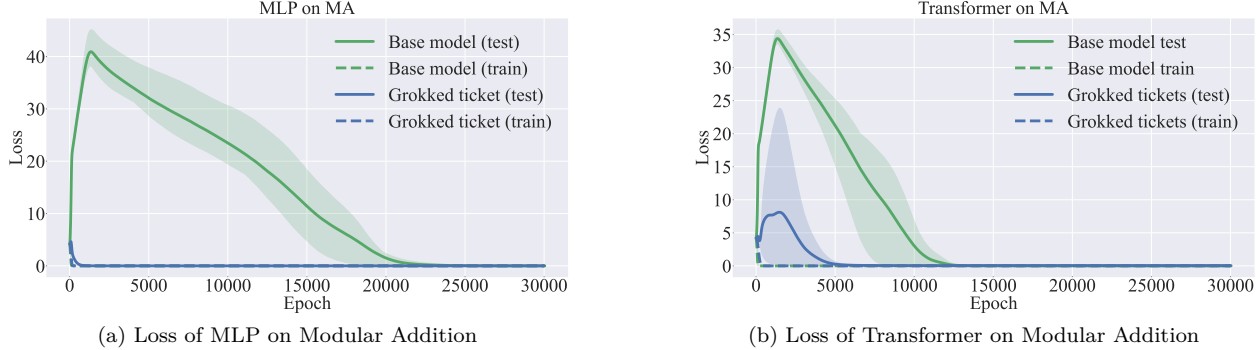

(a) Loss of MLP on Modular Addition

(b) Loss of Transformer on Modular Addition

Figure 11: Train and Test loss of base model and grokked ticket on MLP and Transformer. In both of architectures, grokked ticket convergent 0 faster than base model

Additionally, to ensure reproducibility, we conduct experiments with three different seeds and present the results for experiments in subsection 4.1 and Transformer architecture experiment.

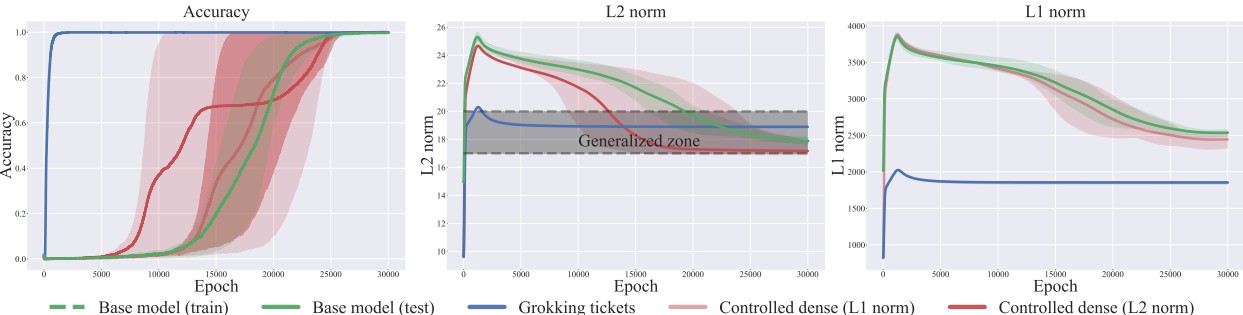

Figure 12: (Left) Test accuracy dynamics of the base model, grokked ticket, and controlled dense model (L1 norm and L2 norm) with three different seeds. The grokked ticket reaches generalization much faster than other models. (Center) L2 norm dynamics of the base model, grokked ticket, and controlled dense model. (Right) L1 norm dynamics of the base model, grokked ticket, and controlled dense models. From the perspective of the L2 norm, all models appear to converge to a similar solution (Generalized zone). However, from the perspective of L1 norms, they converge to different values.

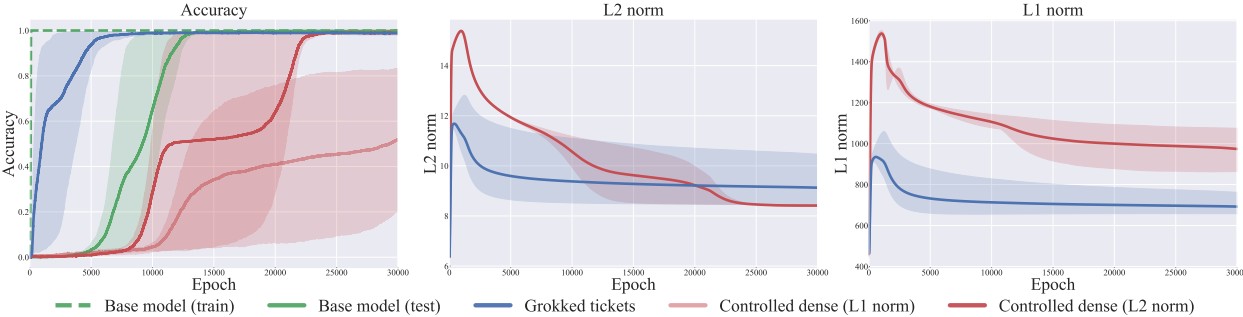

Figure 13: Test accuracy dynamics of the base model, grokked ticket, and controlled dense model (L1 norm and L2 norm) with three different seeds in the Transformer. The grokked ticket reaches generalization much faster than other models. (Center) L2 norm dynamics of the base model, grokked ticket, and controlled dense model. (Right) L1 norm dynamics of the base model, grokked ticket, and controlled dense models. From the perspective of the L2 norm, all models appear to converge to a similar solution (Generalized zone). However, from the perspective of L1 norms, they converge to different values.

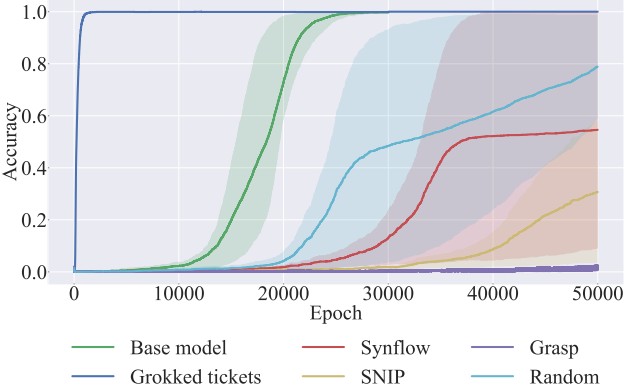

Figure 14: Comparing test accuracy of the different pruning methods. All PaI methods perform worse than the base model or, in some cases, perform worse than the random pruning with three different seeds.

# D  Pruning at initialization methods

Currently, the methodologies of pruning neural networks (NN) at initialization (such as SNIP, GraSP, SynFlow) still exhibit a gap when compared to methods that use post-training information for pruning (like Lottery Ticket). Nonetheless, this area is experiencing a surge in research activity.

The basic flow of the pruning at initialization is as follows:

1. Randomly initialize a neural network $f(\boldsymbol{x}; \boldsymbol{\theta}_0)$.

2. Prune $p\%$ of the parameters in $\boldsymbol{\theta}_0$ according to the scores $S(\boldsymbol{\theta})$, creating a mask m .

3. Train the network from $\boldsymbol{\theta}_0 \odot \boldsymbol{m}$.

According to Tanaka et al. (2020), research on pruning at initialization boils down to the methodology of determining the score in the above process 2, which can be uniformly described as follows:

$$S(\boldsymbol{\theta}) = \frac{\partial R}{\partial \boldsymbol{\theta}} \odot \boldsymbol{\theta}$$

When the $R$ is the training loss $L$, the resulting synaptic saliency metric is equivalent to $|\frac{\partial L}{\partial \boldsymbol{\theta}} \odot \boldsymbol{\theta}|$ used in SNIP (Lee et al., 2019). $-(H\frac{\partial L}{\partial \boldsymbol{\theta}}) \odot \boldsymbol{\theta}$ use in Grasp (Wang et al., 2020).Tanaka et al. (2020) proposed synflow algorithm $R_{SF} = 1^T (\prod_{l=1}^{L} |\boldsymbol{\theta}^{|l|}|)1$. In section 4.2, all initial values were experimented with using the same weights and the same pruning rate.

# E   Significance of Periodicity in the Modular Addition Task

The modular addition task, defined as predicting $c \equiv (a + b) \mod p$, inherently involves periodicity due to the modular arithmetic structure. Nanda et al. (2023) demonstrated that transformers trained on modular addition tasks rely on periodic representations to achieve generalization. Specifically, the networks embed inputs $a$ and $b$ into a Fourier basis, encoding them as sine and cosine components of key frequencies $w_k = \frac{2k\pi}{p}$ for some $k \in \mathbb{N}$. These periodic representations are then combined using trigonometric identities within the network layers to compute the modular sum.

**Mechanism of Periodic Representations**

Nanda et al. (2023) reverse-engineered the weights and activations of a one-layer transformer trained on this task and found that the model computes:

$$\cos(w_k(a+b)) = \cos(w_k a)\cos(w_k b) - \sin(w_k a)\sin(w_k b),$$

$$\sin(w_k(a+b)) = \sin(w_k a)\cos(w_k b) + \cos(w_k a)\sin(w_k b),$$

using the embedding matrix and the attention and MLP layers. The logits for each possible output $c$ are then computed by projecting these values using:

$$\cos(w_k(a+b-c)) = \cos(w_k(a+b))\cos(w_k c) + \sin(w_k(a+b))\sin(w_k c).$$

This approach ensures that the network's output logits exhibit constructive interference at $c \equiv (a+b) \mod p$, while destructive interference suppresses other incorrect values.

Given this mechanism, it can be inferred that the model internally utilizes **periodicity**, such as the addition formulas, to perform modular arithmetic.

# F   Structural Changes in Tasks Other Than the Modular Addition Task

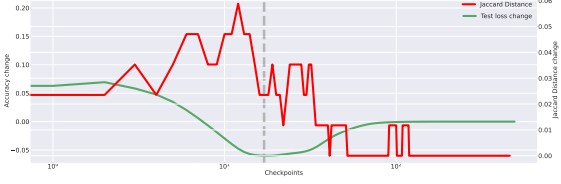
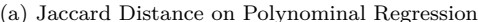
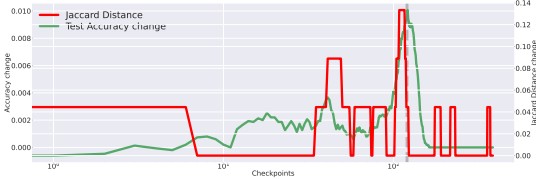

    (a) Jaccard Distance on Polynominal Regression         (b) Jaccard Distance on Sparse Parity

Figure 15: Jaccard distance change and test accuracy change on polynominal regression (a) and sparse parity (b). Structural changes (Jaccard distance) correspond to the acquisition of generalization ability.

## G    Is Weight Norm Sufficient to Explain Grokking in Transformer?

Figure 16 show the accuracy of base model, grokked ticket and Controlled dense (L1 and L2) on Transformer. The results show grokked ticket generalize faster than any other model. The results suggests that even in the case of Transformer, the discovery of grokked ticket is more important than weight norms.

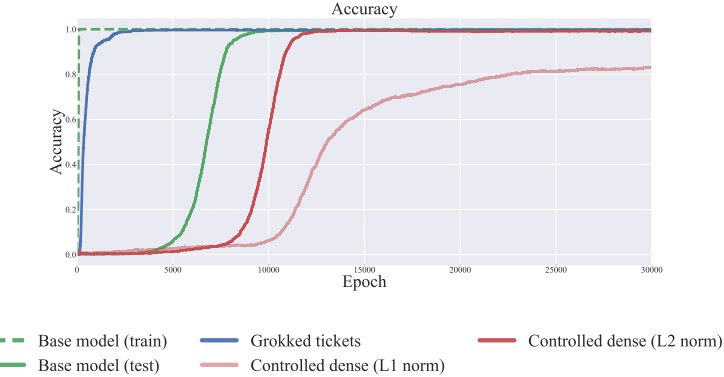

Figure 16: Accuracy of base model, grokked ticket and Controlled dense on Transformer.

## H    Weight Decay work as Structure Explorer

Our result is that the discovery of good structure happens between memorization and generalization, which indicates weight decay is essential for uncovering good structure but becomes redundant after their discovery.

In this section, we first explore the critical pruning ratio, which is the maximum pruning rate that can induce generalization without weight decay Figure 17-(a). We recognize that the critical pruning rate is between 0.8 and 0.9 because if the pruning rate increases to 0.9, the test accuracy dramatically decreases. Thus, we gradually increased the pruning rate in increments of 0.01 from 0.8 and found that the $k = 0.81$ is the critical pruning ratio. We then compare the behavior of the grokked ticket without weight decay and the base model. Figure 17-(b) shows the results of the experiments. The results show that if good structure is discovered, the network fully generalizes without weight decay, indicating that weight decay works as a structure explorer.

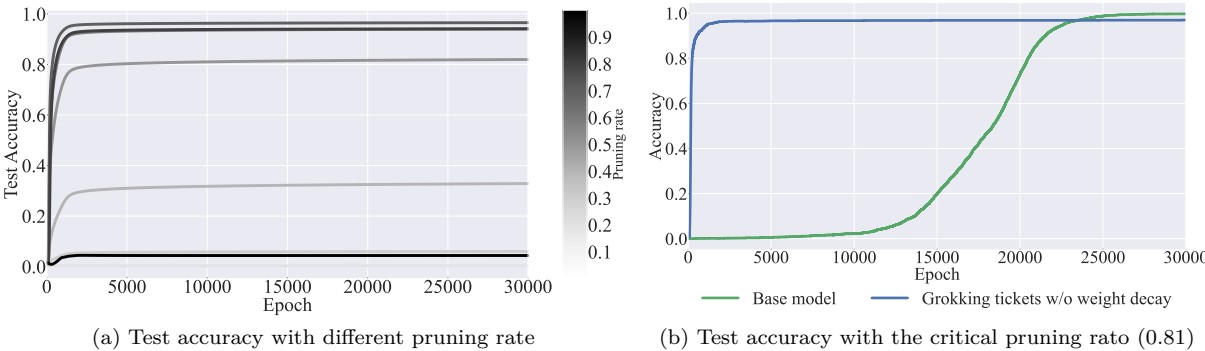

(a) Test accuracy with different pruning rate          (b) Test accuracy with the critical pruning rato (0.81)

Figure 17: The effect of pruning rate on test accuracy without weight decay(left). Test accuracy of grokked ticket with critical pruning rate (0.81) without weight decay(right).

In this section, we show that with precise pruning ratios, the grokked ticket does not require weight decay to generalize, indicating that weight decay is essential for uncovering good subnetworks but becomes redundant after their discovery. We first explore the *critical pruning ratio*, which is the maximum pruning rate that can induce grokking (Figure 17-a). In this case (Figure 17-a), we recognize that the critical pruning rate is

between 0.8 and 0.9 because if the pruning rate increases to 0.9, the test accuracy dramatically decreases. Thus, we gradually increased the pruning rate in increments of 0.01 from 0.8 and found that the $k = 0.81$ is the critical pruning ratio. We then compare the behavior of the grokked ticket without weight decay ($\alpha = 0.0$) and the base model. Figure 17-b show the results of the experiments. As shown in the figure, the test accuracy reaches perfect generalization without weight decay. The results show that the grokked ticket with the critical pruning ratio does not require any weight decay during the optimization.

# I    Visualization of Weights and grokked ticket

To reveal the characteristics of the grokked ticket, we visualized the weight matrices and the corresponding masks of the grokked ticket, as shown in Figure 18. The task under consideration is Modular Addition, implemented using the following MLP architecture:

$$\text{softmax}(\sigma((\boldsymbol{E}_a + \boldsymbol{E}_b)W_{\text{in}})W_{\text{out}}W_{\text{unemb}}), \tag{4}$$

where $\boldsymbol{E}_a$ and $\boldsymbol{E}_b$ are the input embeddings, $W_{\text{in}}$, $W_{\text{out}}$, and $W_{\text{unemb}}$ are the respective weight matrices, and $\sigma$ denotes the activation function. This architecture models the relationship between inputs in the Modular Addition task.

Figure 18 visualizes the learned weight matrices $W_E$, $W_{\text{inproj}}$, $W_{\text{outproj}}$, and $W_U$ (top row) after generalization, as well as the corresponding masks from the grokked ticket (bottom row). The weight matrices exhibit periodic patterns, reflecting good structure learned during training. Furthermore, the grokked ticket masks align with these periodic characteristics, indicating that the grokked ticket has successfully acquired structures that are beneficial for the task of Modular Addition. These results highlight the ability of the grokked ticket to uncover meaningful patterns that contribute to the model's performance.

For comparison, Figure 19 show visualization of masks (structures) obtained by pruning-at-initialization (PaI) methods: Random, GraSP, SNIP, and SynFlow. Unlike the results shown in Figure 18, these methods do *not* exhibit periodic structures. This comparison highlights the superiority of the grokked ticket in acquiring structures that are more conducive to the Modular Addition task, further emphasizing its advantage over traditional PaI methods.

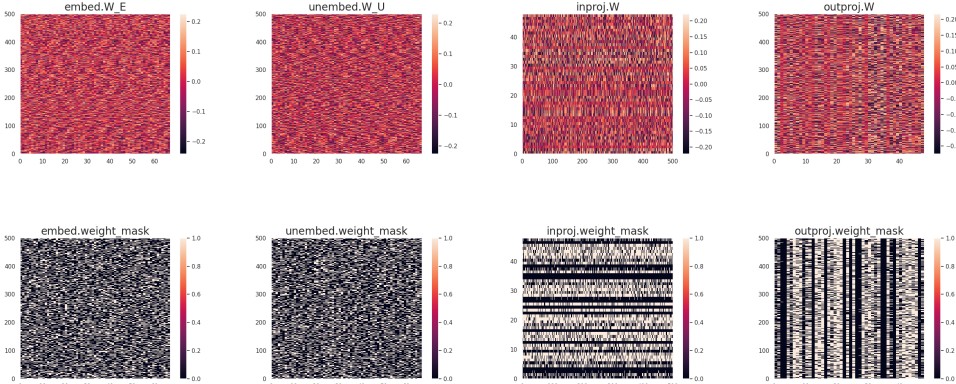

Figure 18: Visualization of weight matrices $W_E$, $W_{\text{inproj}}$, $W_{\text{outproj}}$, and $W_U$ (top), as well as the corresponding masks from the grokked ticket (bottom). Periodic patterns are observed in the weight matrices, and the masks of the grokked ticket reflect these characteristics. This indicates that the grokked ticket has acquired structures beneficial for the task (Modular Addition).

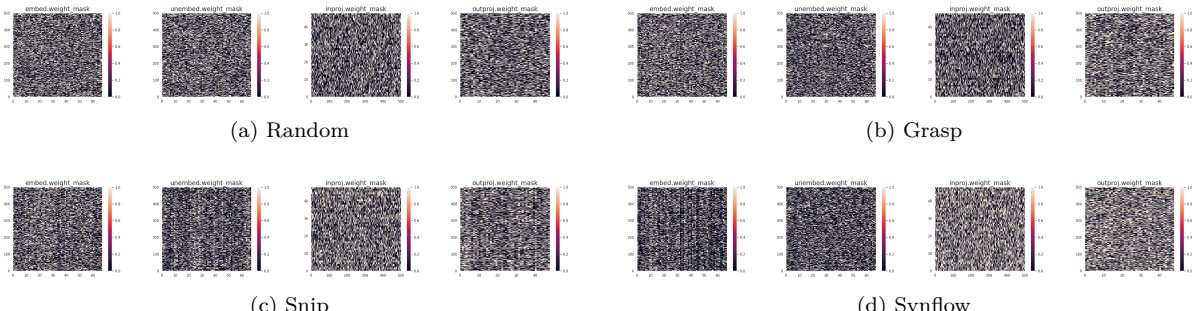

(a) Random

(b) Grasp

(c) Snip

(d) Synflow

Figure 19: Visualization of masks (structures) obtained by pruning-at-initialization (PaI) methods: Random, GraSP, SNIP, and SynFlow. Compared to grokked ticket in Figure 18, it can be observed that periodic structures are *not* achieved.

## J  Graph Property Metrics

In this section, we introduce the graph-theoretic metrics used in our study: **ramanujan gap**, **weighted spectral gap**, and two measures derived from viewing the pruned network as a relational graph — the **average path length** and the **clustering coefficient**.

**Ramanujan Graph.**  A *ramanujan graph* is known for combining sparsity with strong connectivity. Consider a $k$-regular graph, where every node has degree $k$. The eigenvalues of its adjacency matrix $A$ are sorted as

$$\lambda_0 \geq \lambda_1 \geq \lambda_2 \geq \ldots \geq \lambda_n.$$

Here, the largest eigenvalue $\lambda_0$ equals $k$, and the smallest $\lambda_n$ equals $-k$; these are trivial eigenvalues. The graph is called a ramanujan graph if *all non-trivial eigenvalues* $\lambda_i$ (i.e., those with $|\lambda_i| \neq k$) satisfy

$$\max_{|\lambda_i| \neq k} |\lambda_i| \leq \sqrt{2k - 1}.$$

Such graphs are sparse but exhibit efficient information diffusion.

**Ramanujan Gap.**  The ramanujan gap measures how closely a graph's largest non-trivial eigenvalue approaches the theoretical bound $\sqrt{2k-1}$. While this was originally defined for strictly $k$-regular graphs, Hoang et al. (2023b;a) extend it to *irregular graphs* (as often arise in unstructured pruned networks) by replacing $k$ with the average degree $d_{\text{avg}}$:

$$\Delta_r = \sqrt{2\,d_{\text{avg}} - 1} - \hat{\mu}(G),$$

where $\hat{\mu}(G)$ is the largest non-trivial eigenvalue in absolute value. For bipartite graphs, the largest and smallest eigenvalues are symmetric in magnitude, making the third-largest eigenvalue a natural choice for $\hat{\mu}(G)$. Hoang et al. (2023a) reports a negative correlation between $\Delta_r$ and performance.

**Weighted Spectral Gap**  The *spectral gap* of an adjacency matrix typically refers to the difference between its largest eigenvalue $\lambda_0$ and its second-largest eigenvalue $\lambda_1$:

$$\text{Spectral Gap} = \lambda_0 - \lambda_1.$$

Hoang et al. (2023a) extends this to a *weighted spectral gap* by incorporating learned weights (e.g., from training) into the adjacency matrix. This better reflects how pruning or other structural modifications alter the effective connectivity, often revealing strong links to model performance.

Beyond eigenvalue-based measures, interpreting the pruned network as a *relational graph* (You et al., 2020) allows the computation of established network-science metrics:

**Average Path Length.**  The *average path length* $L$ is defined as the mean distance between all pairs of nodes:

$$L = \frac{1}{N(N-1)} \sum_{i \neq j} d(i,j),$$

where $d(i,j)$ is the shortest-path distance between nodes $i$ and $j$, and $N$ is the number of nodes in the graph. A smaller $L$ implies fewer hops on average to traverse between nodes, indicating a more compact network.

**Clustering Coefficient.**  We use the *average clustering coefficient* $C$ to characterize how densely nodes tend to form triangles with their neighbors. The local clustering coefficient $C_i$ of node $i$ is given by

$$C_i = \frac{2e_i}{k_i(k_i - 1)},$$

where $k_i$ is the degree of node $i$ and $e_i$ is the number of edges among its $k_i$ neighbors. The *average clustering coefficient* is then

$$C = \frac{1}{N} \sum_{i=1}^{N} C_i.$$

Higher values of $C$ indicate strong local grouping or community structure.

Interestingly, prior work (You et al., 2020) has shown that when focusing on performance, the best graph structure does not necessarily push *both* average path length and clustering coefficient to their extremes (either too small or too large). Instead, the best-performing networks often occupy a moderate or balanced regime of $L$ and $C$.

## K Evolution of Graph Property for each layer

Figure 20-(a) illustrates how the weighted spectral gap (left) and ramanujan gap (right) evolve throughout training across different layers, specifically analyzing $W_{\text{emb}}, W_{\text{in}}, W_{\text{out}}, W_{\text{unemb}}$.

We observe that the weighted spectral gap exhibits a sharp increase during the memorization phase, indicating a pronounced separation between the largest and second-largest eigenvalues while the model is overfitting. This suggests that, in the memorization phase, weights are dominated by a particular component, corresponding to the leading eigenvector. However, as training progresses and generalization improves, the spectral gap narrows, reflecting a more balanced parameter space utilization. Meanwhile, the ramanujan gap steadily decreases from the beginning, aligning with prior studies (Hoang et al., 2023a) that report a consistent downward trend. This suggests a structural shift in the graph induced by weight matrices, possibly corresponding to the network's shift from memorization to generalization.

Figure 20-(b) presents metrics derived from viewing the entire network as a relational graph (You et al., 2020). The average path length (left) *increases* alongside the rise in test accuracy, then stabilizes at a moderate range. This implies that as the network searches for a generalizable solution (i.e., discovering the Grokked ticket), its effective connectivity becomes more sparse. Once a solution is found, the path length stabilizes, suggesting a structured and efficient parameter allocation. Conversely, the clustering coefficient (right) *decreases* as test accuracy improves, before settling at an intermediate value. This decline suggests that during the discovery of the generalization pattern, local connectivity in the graph weakens, but does not collapse entirely. Instead, it converges to a structured regime where some degree of locality is maintained while avoiding excessive clustering, consistent with previous findings (You et al., 2020).

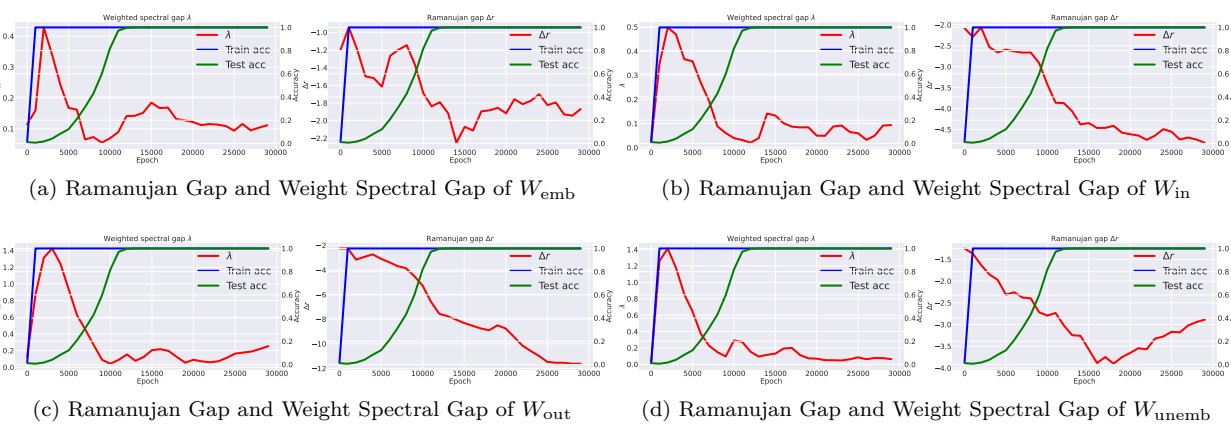

(a) Ramanujan Gap and Weight Spectral Gap of $W_{\text{emb}}$

(b) Ramanujan Gap and Weight Spectral Gap of $W_{\text{in}}$

(c) Ramanujan Gap and Weight Spectral Gap of $W_{\text{out}}$

(d) Ramanujan Gap and Weight Spectral Gap of $W_{\text{unemb}}$

Figure 20: Evolution of the weighted spectral gap and Ramanujan gap of each layer.

