# OpenReview forum: "Bridging Lottery Ticket and Grokking: Understanding Grokking from Inner Structure of Networks"
_TMLR — Accepted by TMLR_

### Review · Reviewer_UuyT · 2025-02-02

**Summary Of Contributions:**

The paper proposes an interesting study about the connection between Grokking tickets and the structure of neural networks. It first shows how sparse networks (discovered by using the Lottery Ticket Hypothesis) can reduce generalization delay with respect to a dense model. Then, the authors empirically demonstrate how both Weight Norm and Sparsity Level are not the main factors that could explain the faster generalization phenomena. The authors then motivate such reduced generalization delay in Grokking tickets based on the Fourier Entropy computed over neurons, showing how Grokking tickets exhibit earlier periodic structure during training. Then they propose a metric based on Jaccard Distance to evaluate the mask change during training and its correlation with task accuracy. The paper concludes by combining a well-known agnostic pruning algorithm (edge-popup) with weight decay to show how the sparse structure discovered by the pruning algorithm improves task performance.

**Audience:**

Yes

**Claims And Evidence:**

No

**Requested Changes:**

I would ask the author to take into consideration my Main Weakness. Even though I liked the idea of the paper and part of its execution (till section 5.2), I missed the link to what a good structure is, which should be one of the main strength points of the paper (along with Sections 4 and 3 and the rejection of Hypothesis 1 and 2).
I think that discovering what a good structure (with some metrics) is is required to support the claim that the network structure matters in grokking tickets. It would be enough to analyze the difference in structures between the network obtained at the beginning of the training and the end with edge-popup. This step, in my opinion, would be a key to strengthening the paper's message. I'm more than willing to discuss this point with the authors during the discussion phase.
Also, check the minors.

**Strengths And Weaknesses:**

## Strengths
* The idea of the paper is novel and smooth; combining delayed generalization and sparse structure is novel.
* The finding that LTH tickets can reduce delayed generalization is novel and interesting.
* The first two findings of the paper (rejection of Hypotheses 1 and 2) are empirically supported.
* The paper is well-written and easy to follow, and the experiments are well-supported (also, all the technical aspects of the experimental setup are included). The appendix and code provide all the details for fair reproducibility.


## Major Weaknesses
* From my perspective, the major limitation is the definition of a good structure throughout the whole paper. No definition is given, and then, in section 5.2, the Jaccard Distance is proposed. However, I do not understand how the JD computed over the binary masks at different epochs can be a measure to define a good structure. It is true that, as shown in Figure 8, JD looks like it’s correlated to the accuracy change; however, this JD says nothing about the structures of the network itself. It just says that it changes. There are several works that analyze the inner structure of sparse neural networks and their correlation with task performance via both basic graph properties [1,2] or expander theory [3,4,5]. In this paper, while I agree with the authors that the structure of the grokking ticket may be the reason that explains such a reduction in generalization delay, I completely miss a metric from the structure of the network that shows me that. Even in the successive experiments with edge-popup, while it’s true that the algorithm changes the binary mask of the sparse network with no weights update, the paper provides no information about the properties of the network at epoch 0 (low accuracy) vs. the last epoch (good accuracy). Understanding the different properties of these two structures (rather than how the structure changes) could lead to revealing why one structure is better than another.

## Minor Weaknesses
* The theory of the Strong Lottery Ticket Hypothesis (hence the mask found by the Edge-Popup algorithm) has been theoretically proven in [6].
* Hypothesis 2: It would be more appropriate to use LTH rather than the PaI algorithm to reject this hypothesis since PaI algorithms are established to be difficult to train from scratch [7]. Hence, the delayed generalization could also be correlated with this phenomenon.
* Some citations are wrongly formatted.
* Appendix E should, in my opinion, be in the main paper.



[1] Stier, Julian, and Michael Granitzer. "Structural analysis of sparse neural networks." Procedia Computer Science 159 (2019): 107-116. \
[2] Cunegatti, Elia, et al. "Understanding Sparse Neural Networks from their Topology via Multipartite Graph Representations." Transactions on Machine Learning Research \
[3] Pal, Bithika, et al. "A study on the ramanujan graph property of winning lottery tickets." International Conference on Machine Learning. PMLR, 2022 \
[4] Hoang, Duc NM, and Shiwei Liu. "Revisiting pruning at initialization through the lens of ramanujan graph." ICLR 2023 (2023). \
[5] Laenen, Steinar. "One-shot neural network pruning via spectral graph sparsification." Topological, Algebraic and Geometric Learning Workshops 2023. PMLR, 2023. \
[6] Malach, Eran, et al. "Proving the lottery ticket hypothesis: Pruning is all you need." International Conference on Machine Learning. PMLR, 2020 \
[7] Evci, Utku, et al. "The difficulty of training sparse neural networks." arXiv preprint arXiv:1906.10732 (2019).

---

> ### Author Response · Authors · 2025-02-21
> **Author Response (1/3)**
>
> We appreciate the thoughtful feedback. We addressed your concerns and changes raised in the review.
>
> We revised the paper based on the reviewers’ comments, and the major edit was highlighted with coloring (**purple**). Please also check the updated manuscript.
>
> **> Major Weakness & Requested Changes**
>
>
> > I do not understand how the JD computed over the binary masks at different epochs can be a measure to define a good structure
> > the paper provides no information about the properties of the network at epoch 0 (low accuracy) vs. the last epoch (good accuracy).
>
> > It would be enough to analyze the difference in structures between the network obtained at the beginning of the training and the end with edge-popup.
>
> Thank you for your insightful comments and constructive suggestions.
> To address your feedback, we have revised our manuscript to provide a clearer explanation of how we define and analyze "good structures" in neural networks.
> Specifically, we have added a paragraph about **Graph Property and Grokked Ticket to Subsection 5.1.**
> Here, following prior work [1,2], we analyze the evolution of the (1) weighted spectral gap, (2) ramanujan gap, (3) average path length, and (4) clustering coefficient of weight matrix  $W_\mathbb{in}$ throughout training, alongside changes in train/test accuracy. Details of each graph property metric are provided in **Appendix J**.
>
> As shown in **Figure 8**, the weighted spectral gap first increases sharply during the memorization phase, indicating that the network’s parameters are dominated by its leading eigenvalue (i.e., a single primary direction). This overfitting state is characterized by a pronounced difference between the largest and second-largest eigenvalues.
> As the model shifts toward generalization, however, the spectral gap narrows, suggesting that the parameter space is utilized more evenly.
> Meanwhile, the ramanujan gap  steadily decreases from the outset of training, which aligns with prior studies [1].
> We observe similar behavior in other layers (see **Appendix K** for details).
>
> We also track the average path length and clustering coefficient in **figure 8**.
> The average path length **increases** alongside the rise in test accuracy, then converges to a moderate range.
> This trend suggests that when the network is searching for a generalized solution (Grokked ticket), its connectivity becomes more sparse; once that solution is found, the path length decreases and stabilizes.
> By contrast, the clustering coefficient **decreases** as test accuracy improves, then also settles into an intermediate value.
> During the phase when the network is searching for a generalized solution (i.e., as the model discovers the Grokked ticket), clustering diminishes but eventually stabilizes near a moderate level.
> Both metrics thus converge to neither extreme but remain in a balanced regime, consistent with prior studies [2].
>
> Additionally, in **Subsection 5.3, Table 1**, we examined the graph properties of $W_\text{in}$ under EP w/o WD.
> As observed in subsection 5.1, the weighted spectral gap initially increased and then declined, while
> the ramanujan gap continued to decrease. Furthermore, both the average path length and the clustering
> coefficient converged to similar values.
>
> We hope these revisions clarify how we define and analyze “good structures” and sufficiently address your comments.
> If you have any additional questions or suggestions, we would be grateful for your continued feedback.
> Thank you again for your thorough and constructive review.
>
> [1] Hoang et.al, “Don’t just prune by magnitude! Your mask topology is a secret weapon”, Neurips 2023.
> [2] You et.al, “Graph Structure of Neural Networks”, ICML2020

---

> > ### Author Response · Authors · 2025-02-21
> > **Author Response (2/3)**
> >
> > **> Minor Weaknesses 1**
> >
> > > The theory of the Strong Lottery Ticket Hypothesis (hence the mask found by the Edge-Popup algorithm) has been theoretically proven in [6]
> >
> >
> > As you pointed out, the theory of the Strong Lottery Ticket Hypothesis (SLTH) itself has already been rigorously established. In our study, we leverage SLTH to empirically investigate our hypothesis that structure discovery occurs between memorization and generalization in the grokking process.
> >
> > Specifically, as shown in Table 1 of our paper, simply applying structural optimization (edge-popup) to the memorized solution is sufficient to achieve generalization, which supports our hypothesis. We emphasize that **our goal is not to experimentally re-verify SLTH.** Instead, we employ **SLTH as a way to highlight how the model transitions from memorization to generalization.**
> >
> > **> Minor Weaknesses 2**
> >
> > > It would be more appropriate to use LTH rather than the PaI algorithm to reject this hypothesis since PaI algorithms are established to be difficult to train from scratch [7]
> >
> > We acknowledge that PaI algorithms are known to be challenging to train from scratch. However, the primary focus of our second hypothesis (Hypothesis 2) is that a “higher degree of sparsity leads to a reduction in delayed generalization.”
> >
> > To test this hypothesis directly, we constructed networks with the same level of sparsity as the grokked ticket. By doing so, we could **isolate the impact of sparsity** and highlight the **unique structural benefits the grokked ticket** may possess beyond mere sparsity alone.
> >
> > **> Minor Weaknesses 3**
> >
> > > Some citations are wrongly formatted.
> > Thank you for your feedback.
> >
> > We have carefully reviewed and corrected the citation formatting in **Subsection 3.1.**
> >
> > **> Minor Weaknesses 4**
> >
> >
> > > Appendix E should, in my opinion, be in the main paper
> >
> > We have incorporated the content from Appendix E into the main paper, specifically in **Subsection 5.1.**
> > This integration ensures that the significance of periodic representations in modular addition is clearly presented, improving the readability and coherence of our discussion on the structure acquired by grokked tickets.

---

> > > ### Author Response · Authors · 2025-02-21
> > > **Author Response (3/3)**
> > >
> > > We appreciate the reviewer’s detailed feedback and thoughtful suggestions. We have carefully addressed the concerns regarding the definition of a good structure by incorporating additional analyses and discussions to strengthen our claims. We hope these revisions clarify our contributions and improve the manuscript. We welcome any further discussion and appreciate the opportunity to refine our work.

---

> ### Comment · Reviewer_UuyT · 2025-02-26
> **Discussion**
>
> Dear Authors,\
> First of all, thanks for taking my suggestions into consideration.
> I just read your response as well as the updated manuscript. I still have some doubts/questions that I would like to discuss with you.
>
> - Why is the structural analysis for both periodic and graph structure only applied to W_in? This, in my opinion, provides information about the structure of only one subset of the network. It would be better to have a single value that represents the structure of the whole network, at least for the Attention and MLP modules.
> - The Ramanujan gap always turns out to be negative, which means that the Ramanujan properties are violated. I don’t know if such a metric suits your case study.
> - I am still worried about the link between JD metrics, graph structure, and performance. The main idea of “the structure matters in grokking,” based on your analysis, lies in the fact that the structure via pruning evolves into a structurally meaningful network. However, while the JD distance increases and the performance on the test set improves, in Table 1, only the Weighted Spectral Gap changes, while the other metrics have the same values during different epochs. This leads me to a question: if you state that EP without weight updates can find the best structure (hence perform well on a given task), which metrics explain the performance improvement with respect to the network structure?
> - If you use only W_in (hence a bipartite directed graph), when computing the average path length, d(I,j) could be either 0 or 1. This metric makes sense if more layers are concatenated together using the relational graph proposed in the original paper. Please correct me if I’m wrong.
> - Table 1 only shows one case of the correlation between structural properties and performance. In order to support this claim, it would probably be better to confirm it in various test cases.
> For EP, which sparsity level have you set? And how does this affect performance?
>
> I would like to express to the authors that I acknowledge the work they put into the revision and that I liked the idea of investigating sparsity in terms of grokking tickets. However, to claim that structure matters, it is important to find more stable metrics—in plain words, to understand which metrics change during the training phase of EP without weight decay in order to isolate the structure alone. At the current stage, out of four metrics, only the Weighted Spectral Gap changes its values during training. However, it is the only metric that relies on the structural weight values, while the weight value-agnostic metrics (path length and clustering coefficient) provide no information since they are almost fixed during training.

---

> > ### Author Response · Authors · 2025-03-03
> > **Author Reply**
> >
> > We sincerely appreciate the time and effort the reviewer has taken to carefully read our revised manuscript and provide thoughtful feedback. Your insights have been invaluable in refining our work, and we truly appreciate your constructive criticism and detailed questions. Below, we address the concerns raised in your review.
> >
> > **> W1**
> > >  Why is the structural analysis for both periodic and graph structure only applied to W_in?
> >
> > Our main text primarily focuses on $W_\text{in}$​, but in **Appendix K** we also provide results for $W_\text{emb}$​, $W_\text{out}​$, and $W_\text{enmeb}$. All of these exhibit the same overall trends, so please refer to those as well.
> >
> > We agree that summarizing the structural properties of the entire network with a single value would be beneficial. However, given that each layer contributes differently to the network’s functionality, analyzing them individually allows us to capture layer-specific patterns. In the context of grokking for Modular Addition, it has been shown that each layer in the network corresponds to a specific algorithm [1,2].
> > Although we found that all layers exhibit similar graph properties, we decided to present separate plots for each layer in order to highlight the relationship with prior works.
> >
> > [1] Nanda et al., “Progress measures for grokking via mechanistic interpretability”, ICLR2023.
> > [2] Zhong et al., “The Clock and the Pizza: Two Stories in Mechanistic Explanation of Neural Networks”, Neurips 2023
> >
> > **> W2**
> > > The Ramanujan gap always turns out to be negative, which means that the Ramanujan properties are violated. I don’t know if such a metric suits your case study.
> >
> > As you rightly point out, we observe negative Ramanujan gaps. However, this is consistent with previous findings in ResNet [3], where negative values were also associated with improved test performance. These results suggest that, even though the gap is negative, it remains a meaningful indicator in our context.
> >
> > [3] Hoang et al., ”Don’t just prune by magnitude! Your mask topology is a secret weapon”, Neurips 2023
> >
> > **> W3**
> > > If you use only W_in (hence a bipartite directed graph), when computing the average path length, d(I,j) could be either 0 or 1.
> >
> > There may have been some misunderstanding regarding the interpretation of the path length and clustering coefficient results in **Figure 8-(b).**
> > As noted in the main text, the path length and clustering coefficient results shown in **Figure 8-(b)** come from treating the **entire network as a relational graph**, not just $W_{\text{in}}$​. Therefore, $d(i,j)$can take values beyond 0 or 1.
> >
> > To clarify this point, we have emphasized the term **relational graph** in the paper.
> >
> > **> W4**
> >
> > > However, while the JD distance increases and the performance on the test set improves, in Table 1, only the Weighted Spectral Gap changes, while the other metrics have the same values during different epochs.
> >
> > > the weight value-agnostic metrics (path length and clustering coefficient) provide no information since they are almost fixed during training.
> >
> > The observation that only the Weighted Spectral Gap appears to change is primarily due to differences in **scale**. As shown in **Figure 8-(b)**, both the average path length and the clustering coefficient also exhibit noticeable variations in relation to test accuracy. The results in **Table 1** are consistent with those in **Figure 8**, demonstrating that as test accuracy improves, the Weighted Spectral Gap decreases, and the Ramanujan Gap also declines. Similarly, the average path length and clustering coefficient follow the same trend as the other metrics, decreasing over time.
> >
> > Moreover, to address your concern more comprehensively, we have added the results for the **Spectral Gap** (not the Weighted Spectral Gap) in **Table 1**. The Spectral Gap is a **weight-value-agnostic** graph property metric, and our results indicate that it also decreases as test performance improves. This further supports our central claim that structural properties play a crucial role in the grokking phenomenon.
> >
> > We appreciate your feedback, as it helps reinforce the significance of our findings.
> >
> > **> W5**
> >
> > > For EP, which sparsity level have you set? And how does this affect performance?
> >
> > As stated in Section 3.1, we use a default pruning rate of 0.6. While we have not yet conducted experiments with other pruning rates, we plan to do so and will report the results as soon as they become available. Thank you for your valuable suggestion.
> >
> >
> > We sincerely appreciate your thoughtful feedback, which has helped us refine our work. If you have any further questions or suggestions, we would be happy to address them.

---

> ### Comment · Reviewer_UuyT · 2025-03-04
>
> Dear Authors, \
> Thanks for the further explanation.
>
> I believe your response addresses some of my concerns. However, as a minor point, I think it would be better to explicitly state in the caption of Table 8 that you used a relational graph for the average path length and clustering coefficient. Concerning Table 1, the caption states, “we report graph property metrics of the network’s W_in matrix.” Hence, it is unclear whether the graph metrics reflect the relational graph (for average path length and clustering coefficient) or the bipartite analysis over W_in for the Ramanujan metrics. For the Ramanujan metrics, did you average them over all the layers of the model as in the original paper?
>
> Apart from this minor issue, even though I appreciate and value the authors’ effort during the rebuttal, my main concern still holds. I will try to express my point. The paper is sound and interesting until Section 5, and the strengths I mentioned in my first review still hold. The investigation of grokking and sparsity is novel, promising, and interesting. However, Section 5 is still too confusing, and more experiments and analyses are required, in my perspective, to make the submission exceed the acceptance bar. The main issues at the moment are the following:
>
> - Some metrics are shown by layer (Ramanujan ones), while others, such as average path length and clustering coefficient, are computed for the whole graph using a different graph encoding (relational graph). This is somewhat confusing for readers.
> - Table 1 only shows the correlation of metrics with performance over test accuracy in one experimental scenario. While I don’t think large-scale experiments are needed for this paper, I believe that such an analysis should be conducted on more tasks (modular arithmetic, polynomial regression, sparse parity, and MNIST classification) and different models to claim a pattern between the sparse structure and performance. Additionally, different models and seeds need to be used to validate the pattern.
> - JD distance should be better linked to metrics. Currently, Figure 9 only correlates it with test accuracy change.
> - If the statement of the paper is “the structure matters, and with edge-pop we can achieve good performance in a training-free setting,” then you should also be able to provide details about such a structure and possibly a way to sample it.
>
> To conclude, while I believe the paper has its strengths, in my opinion, after Section 5, the weaknesses outweigh the strengths. In particular, a more detailed analysis of the correlation between structural properties and task performance is needed to support the paper’s claims.
>
> I would also highlight that I acknowledge that the other reviewers champion the paper; hence, I will not strongly oppose accepting it. However, these are my final considerations about the submission, and I hope they can be taken into account by the authors in order to strengthen the paper.

---

> > ### Author Response · Authors · 2025-03-10
> > **Author Reply  (1/2)**
> >
> > Thank you for your continued engagement and thoughtful feedback. We truly appreciate the time and effort you have invested in reviewing our work and providing such detailed comments. We have updated the paper and marked all revisions in purple for clarity.
> >
> > **> W1**
> >
> > > Some metrics are shown by layer (Ramanujan ones), while others, such as average path length and clustering coefficient, are computed for the whole graph using a different graph encoding (relational graph). This is somewhat confusing for readers.
> >
> > Thank you for pointing out this potential confusion. Following your suggestion, we now compute both the weighted spectral gap and the Ramanujan gap as averages across all layers, thereby treating them as single metrics for the entire network. In **Figure 9**, we show that these averaged metrics exhibit the same overall trends as those observed for $W_\text{in}$ . To maintain clarity and completeness, we include the per-layer metrics in **Appendix K**.
> >
> > We hope this addresses your concern regarding the clarity of these metrics.
> >
> > **> Minor Weakness**
> >
> > > For the Ramanujan metrics, did you average them over all the layers of the model as in the original paper?
> >
> > Thank you for noting the ambiguity.
> > In the same manner as **W1**, we compute these Ramanujan metrics by averaging the graph property metrics across all layers.
> >
> > **> W2**
> >
> > > Table 1 only shows the correlation of metrics with performance over test accuracy in one experimental scenario.
> >
> > We will run additional experiments with multiple random seeds during the rebuttal period to verify the consistency of our findings. We will share these new results as soon as they are available, and we hope this more comprehensive analysis will address your concern.
> >
> > **> W3**
> >
> > > JD distance should be better linked to metrics. Currently, Figure 9 only correlates it with test accuracy change.
> >
> > > To conclude, while I believe the paper has its strengths, in my opinion, after Section 5, the weaknesses outweigh the strengths.
> >
> > Thank you for your suggestion. In order to address your concern, we have reorganized the structure of Section 5’s subsections as follows (specifically, by swapping the order of the original Subsections 5.1 and 5.2):
> >
> > - **Subsection 5.1: Progress Measure: Structural Shift Captures the Timing of Generalization**
> >   Here, we first demonstrate that changes in the network’s structure coincide with improvements in test accuracy.
> >
> > - **Subsection 5.2: Analysis of Good Structures Through Periodic Structures and Graph Properties**
> >   At the beginning of this subsection, we provide the following explanation to connect our analysis of JD distance with graph metrics:
> >   > In the previous subsection (subsection 5.1), we established that significant changes in the network’s structure, as measured by Jaccard Distance, coincide precisely with improvements in test accuracy. Building on this finding, we now delve deeper into the structural properties that trigger such abrupt differences. Specifically,
> > we investigate the nature of the discovered structure from two different perspectives: (1) the periodic
> > representations known to emerge in modular arithmetic tasks (as studied by Pearce et al. (2023); Nanda et al.
> > (2023)), and (2) graph-theoretic properties that reveal how the network’s connectivity evolves to support
> > better generalization. By examining both viewpoints, we uncover how the network ultimately settles on a
> > ‘good structure’ that drives high test accuracy
> >
> > - **Subsection 5.3: Pruning during Training: Pruning Promotes Generalization**
> >
> > We believe these changes address your concern in **Section 5** regarding the correlation between JD distance, structural properties, and task performance.

---

> > > ### Author Response · Authors · 2025-03-10
> > > **Author Reply (2/2)**
> > >
> > > **> W4**
> > >
> > > > If the statement of the paper is “the structure matters, and with edge-pop we can achieve good performance in a training-free setting,” then you should also be able to provide details about such a structure and possibly a way to sample it.
> > >
> > > As you pointed out, there are indeed previous studies [1] suggesting that networks characterized by low average path lengths and high clustering coefficients can enhance learning efficiency.
> > > Motivated by your initial review comments, we added Figure 9 and Table 1 during the rebuttal period, identifying generalized subnetworks characterized by a low spectral gap, low Ramanujan gap, and average path length and clustering coefficient properties that evolve distinctively during training, which are beneficial for learning. While we recognize the importance of explicitly sampling architectures based on such graph properties, empirically validating improved learning efficiency through structural sampling remains beyond the scope of the current study. We have clearly identified this as an important future direction and included it in **Section 6.**
> > > Specifically, we have added the following discussion:
> > > > Additionally, based on the insights regarding graph properties obtained in subsection 5.2 of this study, future work should explicitly sample neural network structures according to graph properties, as demonstrated in previous research (Javaheripi et al., 2021). Incorporating the structural insights identified here into the sampling process could facilitate the discovery of more efficient and robust network architectures.
> > >
> > >
> > > Once again, we sincerely appreciate your valuable feedback and welcome any further comments or concerns you may have.
> > >
> > > [1] M. Javaheripi, B. D. Rouhani and F. Koushanfar, "SWANN: Small-World Architecture for Fast Convergence of Neural Networks," in IEEE Journal on Emerging and Selected Topics in Circuits and Systems.

---

> > > > ### Author Response · Authors · 2025-03-18
> > > >
> > > > We also included the mean and standard error across multiple seeds in Table 1. Below are the results. We found that even when changing the random seed, the trends of the five graph property metrics (weighted spectral gap, spectral gap, ramanujan gap, average path length, and clustering coefficient) remained consistent.
> > > >
> > > > We believe that these results further strengthen the validity of our paper.
> > > >
> > > > | **Epoch** | **600**           | **1000**          | **1400**          | **2000**          |
> > > > |-----------|--------------------|-------------------|-------------------|-------------------|
> > > > | **Test Accuracy (\%)**       |                    |                   |                   |                   |
> > > > | WD                         | 0.53 ± 0.31       | 0.95 ± 0.03      | 1.00 ± 0.00      | 1.00 ± 0.00      |
> > > > | EP w/o WD                  | 0.68 ± 0.19       | 0.80 ± 0.17      | 0.84 ± 0.16      | 0.92 ± 0.06      |
> > > > | EP w/ WD                   | **0.82 ± 0.04**   | **0.96 ± 0.01**  | 0.99 ± 0.00      | 1.00 ± 0.00      |
> > > > | **Graph Property Metrics**    |                    |                   |                   |                   |
> > > > | Weighted spectral gap      | 0.612 ± 0.05      | 0.810 ± 0.11     | 0.479 ± 0.04     | 0.358 ± 0.03     |
> > > > | Spectral gap               | 48.91 ± 5.05      | 48.51 ± 5.12     | 47.42 ± 4.97     | 47.20 ± 4.55     |
> > > > | Ramanujan gap              | -2.550 ± 0.52     | -4.022 ± 0.05    | -4.335 ± 0.02    | -4.555 ± 0.01    |
> > > > | Average path length        | 1.964 ± 0.006     | 1.961 ± 0.002    | 1.960 ± 0.001    | 1.959 ± 0.001    |
> > > > | Clustering coefficient     | 0.733 ± 0.41      | 0.724 ± 0.35     | 0.719 ± 0.30     | 0.719 ± 0.28     |

---

### Review · Reviewer_3B6m · 2025-02-05

**Summary Of Contributions:**

This paper works to unite the study two of machine learning's spookiest phenomena: lottery tickets and grokking. The key contribution of the paper is showing that the network structure, as studied through the lens of lottery tickets, can help to explain grokking better than sparsity or norms. The paper's central experiment is that generalization speeds up when you use lottery ticket pruning. They also do work to disentangle hypotheses about whether weight norms, sparsity, or lottery tickets explain grokking, and they show that norm and sparsity alone are insufficient (but still seem helpful).

Overall, I think that this paper is well done and interesting, though not clearly impactful. I currently lean toward acceptance with minimal revisions.

**Audience:**

Yes

**Claims And Evidence:**

Yes

**Requested Changes:**

Consider adding answers to the questions asked at the top of section 5.

So what? I think it would be good to add a paragraph in the discussion to discuss future work and/or what we should do with these results.  I don't know if there's much of a "so what," but feel free to discuss what there might be.

**Strengths And Weaknesses:**

S1: I am not the biggest expert in lottery tickets and grokking, but, like many people, have read a good bit about them in the past few years, this seems like the most illuminating one I have seen.

S2: Writing is good and clear. For example, outlining the questions at the top of section 5 is nice. Feel free to answer them up there too.

S3: I think that the experiments have been done thoroughly. For example, I think this paper would have been viable missing one or two of the subsections of section 5. But the authors did all 3.

W1: As is a general concern with lottery ticket/grokking work, it is not clear whether this is useful. Experiments focus on basically understanding lottery tickets and grokking but don't suggest any clear type of value to practitioners. This is ok, and it's still valid science. But it would be nice to start seeing the extent to which these insights help anyone. Grokking and lottery tickets are popular topics, and I'm not penalizing the paper for working on them, but research on them increasingly seems like the self-perpetuating pursuit of some curiosities.

W2: Experiments are pretty small scale. But this is a problem of the lottery ticket/grokking literature at large. So I don't fault this paper for simple tasks/nets. However, it would be nice if experiments found implications for more sota models.

---

> ### Author Response · Authors · 2025-02-21
> **Author Response (1/2)**
>
> We appreciate the thoughtful feedback. We addressed your concerns and changes raised in the review.
>
> We revised the paper based on the reviewers’ comments, and the major edit was highlighted with coloring **(purple)**. Please also check the updated manuscript.
>
> **> W1 & R2**
>
> > Experiments focus on basically understanding lottery tickets and grokking but don't suggest any clear type of value to practitioners.
>
> > I think it would be good to add a paragraph in the discussion to discuss future work and/or what we should do with these results
>
> We appreciate the reviewer’s question regarding the practical significance of our work and its potential value to practitioners.
> Our findings that good subnetworks (i.e., “lottery tickets”) play a pivotal role in mitigating delayed generalization offer a new perspective on **regularization** and have practical implications:
>
> Traditional regularization techniques, such as weight decay, focus on constraining the magnitude of the network’s parameters. However, our results imply that effective regularization should encourage **the discovery of beneficial structures** within the network. From this viewpoint, weight decay can be viewed as an indirect driver of structural exploration.
> By contrast, explicitly searching for good subnetworks, such as via pruning methods or specialized mask-optimization algorithms, holds the potential for more direct and potentially more effective regularization strategies.
>
> For example, table 1 of our paper demonstrates that supplementing weight decay with the edge-popup algorithm significantly accelerates the transition from memorization to generalization.
> This underscores that approaches optimizing structural properties (e.g., subnetworks) can amplify or surpass the benefits of traditional weight-norm constraints alone.
>
> These insights suggest that practitioners may improve generalization by incorporating methods that directly optimize beneficial structures rather than solely relying on traditional regularization techniques like weight decay. Our results pave the way for developing new, **structure-oriented regularization** techniques to better leverage the benefits of grokking tickets in practical applications.
>
> To make this point clearer, we have added a new paragraph, **Future Directions (Section 6)**, where we discuss the practical significance of our findings and how they might be implemented in real-world scenarios.
>
> **> W2**
>
> > it would be nice if experiments found implications for more sota models.
>
> We acknowledge the reviewer’s point that our experiments are small-scale. However, our primary goal was to investigate grokking in the standard settings established by prior studies (e.g., multiple modular arithmetic operations, polynomial regression, sparse parity, and MNIST classification) using both MLP and Transformer architectures. These tasks and models are consistent with most existing grokking literature, providing a controlled environment to validate our hypotheses.
>
> We also agree that exploring larger models, such as large language models (LLMs), and more complex, real-world datasets would offer valuable insights into the practical impact of our results. As noted in our newly added **Future Directions (Section 6)**, we plan to extend our experiments to larger models and real-world datasets to assess the applicability of our insights to state-of-the-art scenarios.

---

> > ### Author Response · Authors · 2025-02-21
> > **Author Response (2/2)**
> >
> > **> R1**
> >
> > > Consider adding answers to the questions asked at the top of section 5.
> >
> > In response to your suggestion, we have explicitly addressed the key questions posed at the beginning of **Section 5** by adding clear answers at **the end of Subsections 5.1, 5.2, and 5.3.** Below are the added Q&A pairs:
> >
> >
> > **Q1**: What exactly constitutes a good structure?
> >
> > **A1**: A good structure has low Fourier Entropy, indicating periodic weight patterns aligned with the task. It also
> > follows known graph properties, such as a rising weight spectral gap during memorization and an increasing
> > average path length in generalization
> >
> > **Q2**: Is the acquisition of structure synchronized with the improvement in generalization performance?
> >
> > **A2**: The discovery of a good structure, as measured by Jaccard Distance, occurs simultaneously with test accuracy improvements, indicating that structural changes drive generalization.
> >
> > **Q3**: Can generalization be achieved solely through structural exploration (pruning) without weight updates?
> >
> > **A3**: Pruning alone significantly improves generalization, demonstrating that discovering good subnetworks is sufficient to transition from memorization to generalization.
> >
> >
> >
> > By incorporating these explicit answers, we have clarified the core takeaways of our study, making the findings more accessible and actionable.
> >
> > Thank you for your valuable feedback.
> > We believe these clarifications and revisions have strengthened the manuscript.

---

> > > ### Comment · Reviewer_3B6m · 2025-02-21
> > > **Thanks. I think the paper should be accepted.**
> > >
> > > Thanks for the reply. I appreciate the additions mentioned, and I have no additional questions. Congrats to the authors on a good paper. Unless any other reviewers have major concerns, I think this would be a good one to accept.

---

> > > > ### Author Response · Authors · 2025-03-03
> > > >
> > > > Thank you for reviewing our response. If you have any additional questions, we would be happy to address them within the remaining timeline.
> > > >
> > > > We sincerely appreciate your engagement and effort given the tight schedule.

---

### Review · Reviewer_dmLd · 2025-02-25

**Summary Of Contributions:**

This paper investigates the phenomenon of grokking, where neural networks transition from memorization to generalization. The authors argue there is a connection between grokking and the Lottery Ticket, in particular, grokking has more to do with good subnetworks than weight norm and sparsity. The research demonstrates that lottery tickets obtained at the generalization phase significantly reduce delayed generalization across various tasks such as modular arithmetic, polynomial regression, sparse parity, and MNIST classification.

**Audience:**

Yes

**Claims And Evidence:**

Yes

**Requested Changes:**

Please address weaknesses.

**Strengths And Weaknesses:**

## Strengthes

1. The paper makes a novel, interesting connection between grokking and the Lottery Ticket, which seems to me a new perspective on delayed generalization.

2. Authors have perform comprehensive experiments to validate their hypothesis, that is, it is the good structures that are responsible for the fast convergence rather than reduce weight and increased sparsity.

3. Authors have done some qualitative analysis to shed light on what constitute a good substructure. Finally, authors demonstrated that pruning alone can transition the network from memorizing to generalizing solutions.

## Weaknesses

1. Limited evaluation setting. Would the results generalize to real-world applications, such as in the large, complex vision or language models?

2. For the pruning method presented in 5.3, the authors state that:

> To verify this, we introduce edge-popup (Ramanujan et al., 2020), a method that learns how to prune
weights without weight updates

Why this particular method, why not others? Would the results differ significantly if switched to other pruning methods?

---

> ### Author Response · Authors · 2025-03-03
> **Author Response**
>
> We appreciate the thoughtful feedback. We addressed your concerns and changes raised in the review.
>
> We revised the paper based on the reviewers’ comments, and the major edit was highlighted with coloring (**purple**). Please also check the updated manuscript.
>
> **> W1**
>
> > Limited evaluation setting. Would the results generalize to real-world applications, such as in the large, complex vision or language models?
>
> We agree that our experiments, while covering multiple modular arithmetic operations, polynomial regression, sparse parity, and MNIST classification — tasks commonly used in the grokking literature — have not yet been extended to large-scale models for real-world applications.
>
> We have made this limitation explicit in the revised **Section 6 (Future Directions)**, which now emphasizes the importance of extending these findings to more complex and large-scale models (e.g., in vision and language). This next step will help validate and refine our approach in real-world applications.
>
> **> W2**
>
> > Why this particular method, why not others? Would the results differ significantly if switched to other pruning methods?
>
> We used the edge-popup algorithm, which is well known as a method for optimizing networks without updating weights. This approach is also known as the Strong Lottery Ticket Hypothesis.
>
> The differences between this method and other pruning techniques can be seen in **Figure 5-(b).**
> In our experiments, we tested three well-known "pruning at initialization" methods—SNIP, GraSP, and SynFlow. However, none of them improved generalization performance. This suggests that certain pruning methods may actually degrade generalization performance.
>
> To clarify this point, we have revised the **Regularization paragraph in Section 6,** explicitly stating that some pruning methods can negatively impact generalization performance.
>
> Thank you for your valuable feedback. We believe these clarifications and revisions have strengthened the manuscript.

---

### Author Response · Authors · 2025-03-18
**Summary of Discussion**

Dear Reviewers and Action Editor,

We would like to thank you for the time and effort you have invested in reviewing our paper. As the discussion period draws to a close, we believe our responses have thoroughly addressed reviewers’ concerns. We would like to highlight several important points from the discussion and our contributions, and we hope you will take these into consideration when making your final decision.

## **> Discussion and Limitation**

Throughout the discussion period, we updated **Section 6** to address the reviewers’ comments more thoroughly:

- **Regularization paragraph**: We clarified that pruning methods other than edge-popup may degrade test performance, while edge-popup can improve generalization (from **dmLd**).
- **Future Direction paragraph**: We emphasized the practical implications of our findings for real-world applications. In particular, we proposed that structure-oriented regularization may outperform conventional approaches (e.g., weight decay), since our results suggest that discovering a “good structure” is more critical than simply reducing weight norms (from **3B6m**).
- **Future Direction paragraph**: We also discussed a graph-property-based structural sampling method, informed by the results in **Figure 9** and **Table 1**. We suggest that an efficient structural sampling approach could be a promising direction for future research. (from **UuyT**)

We believe these updates address the reviewers’ concerns and more clearly outline the limitations and future directions of our work.

## **> Graph Property Experiments**

During the discussion, we added experiments drawing on prior research into graph properties in neural networks, further supporting our central claim that discovering beneficial structures within the network is key to the “grokking” phenomenon (from **UuyT**). Specifically:

- **Figure 9** shows how four metrics (weighted spectral gap, Ramanujan gap, average path length, and clustering coefficient) evolve over training, alongside corresponding changes in training/test performance.
- **Appendix K** includes detailed results for these four metrics across each layer.
- **Table 1** provides the graph property metrics when using the edge-popup pruning algorithm.

We believe these follow-up experiments on network structural indicators help address the major question of what constitutes a “good structure.” By illustrating how specific graph properties change during training, we strengthen our argument that identifying beneficial network structures is crucial for enhanced performance.

## **> Positioning of Section 5 of Our Paper**
We have also revised **Section 5** for greater clarity regarding its role in the paper:

- We explicitly answer the key questions posed at the beginning of Section 5 at the end of Subsections 5.1, 5.2, and 5.3 (from **3B6m**).
- To clarify the relationship among JD distance, structural properties, and task performance, we reorganized Section 5 by swapping the original Subsections 5.1 and 5.2. We also added an explanation at the beginning of Subsection 5.2, connecting our JD distance analysis to the graph metrics (from **UuyT**).

We believe these revisions clarify the purpose of Section 5 and clearly show how our findings contribute to the overall argument.

Sincerely, Authors

---

### Decision · Action_Editor_GU98 · 2025-04-09

**Recommendation:** Accept as is

**Comment:**

This paper investigates the mechanism of grokking, a delayed generalization phenomenon in which neural networks transition from memorization to generalization. Specifically, the paper connects grokking to the lottery ticket hypothesis (LTH), experimentally demonstrating that the network structure has a greater impact on grokking than previously known factors such as weight norm and sparsity.

The paper has clear strengths and weaknesses.

Strengths:

1). Paper is clear written and easy to follow.

2). The paper establishes an innovative connection between grokking and LTH, offering new insights into the mechanisms behind delayed generalization and generalization of neural networks.

3). The paper provides strong evidence challenging previous hypotheses, i.e. that weight norm and sparsity are the main factors influencing grokking, by highlighting the importance of network structure in achieving efficient generalization.

4). During rebuttal, the authors provide detailed clarifications and additional experimental results.

Weaknesses:

1). The evidence and explanation regarding the good structure are somewhat indirect and empirical.

2). The paper lacks large-scale demonstrations and does not provide clear discussion on practical implications.

Despite these weaknesses, the paper opens promising new directions for understanding delayed generalization and generalization of neural networks, which could be of interest to some TMLR’s audience. Therefore, I recommend acceptance.

**Audience:**

Yes, particularly for researchers working in the area of delayed generalization and generalization mechanisms of neural networks.

**Claims And Evidence:**

Yes